# Towards Ultra-High-Definition Image Deraining: A Benchmark and An Efficient Method

## Abstract

Despite significant progress has been made in image deraining, existing approaches are mostly carried out on low-resolution images. The effectiveness of these methods on high-resolution images is still unknown, especially for ultra-high-definition (UHD) images, given the continuous advancement of imaging devices. In this paper, we focus on the task of UHD image deraining, and contribute the first large-scale UHD image deraining dataset, 4K-Rain13k, that contains 13,000 image pairs at 4K resolution. Based on this dataset, we conduct a benchmark study on existing methods for processing UHD images. Furthermore, we develop an effective and efficient architecture (called UDR-Mixer) to better solve this task. Specifically, our method contains two building components: a spatial feature rearrangement layer that captures long-range information of UHD images, and a frequency feature modulation layer that facilitates high-quality UHD image reconstruction. Extensive experimental results demonstrate that our method performs favorably against the state-of-the-art approaches while maintaining a lower model complexity. The code and dataset will be available to the public.

## 1 Introduction

Single image deraining aims to remove the undesired degradation induced by rain streaks from input images, enhancing its visual quality and improving the accuracy of perception system (Chen et al., 2022b). In image deraininig, deep learning-based methods become predominate ones as the formation of image deraining is quite simplified compared to the conventional prior-based methods (Luo et al., 2015). One can choose deep models based on convolutional neural network (CNN) (Jiang et al., 2020; Zamir et al., 2021; Yi et al., 2021) or Transformer architectures (Xiao et al., 2022; Chen et al., 2023b;a; 2024) to directly estimate clear image from rainy one.

Among these approaches, most of them are trained and evaluated on low-resolution datasets (Chen et al., 2023c). These commonly used benchmark datasets consist of 1K or even lower resolution images, such as Rain200L/H (Yang et al., 2017) and Rain13k (Jiang et al., 2020), as illustrated in Figure 1(a). Based on the existing empirical studies (Zhang et al., 2021; Wang et al., 2023; Li et al., 2023), existing image deraining approaches trained on these low-resolution datasets are not likely to generalize well on high-resolution images. However, few effort has been made in ultra-high definition (UHD) image deraining due to the absence of UHD deraining dataset. As UHD devices have been widely used, it is urgent and essential to build a high-resolution benchmark and pave the way for future research in this field.

To explore the performance of existing approaches on UHD images, in this paper, we first establish a large-scale dataset called 4K-Rain13k to benchmark existing methods. The proposed 4K-Rain13k contains 13,000 rainy/rain-free image pairs at 4K resolution ($3840 \times 2160$), with 12,500 pairs allocated for training and 500 pairs for testing. Unlike existing datasets (Yang et al., 2017; Fu et al., 2017; Zhang & Patel, 2018; Zhang et al., 2019; Jiang et al., 2020; Liu et al., 2021b) that directly add rain streaks proportionally to clear images to synthesize rainy images, we observe that geometric inconsistencies in the lengths and thicknesses of rain streaks between low-resolution and high-resolution rainy images. In high-resolution UHD images, rain streaks typically appear longer and slighter due to the increased pixel information, whereas in low-resolution images, the same length of rain streaks may appear blurred into shorter or thicker lines due to fewer pixels. To this end, by integrating geometric transformations (i.e., scaling) into the rain synthesis pipeline, we can enhance

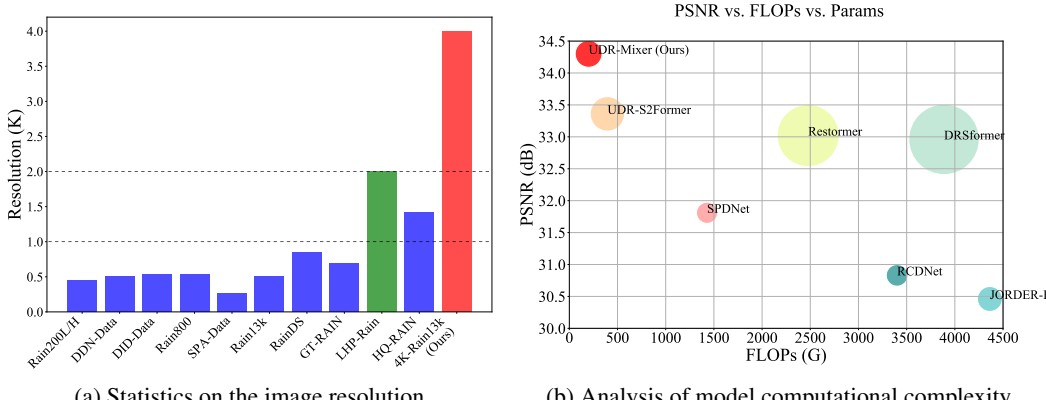

(a) Statistics on the image resolution    (b) Analysis of model computational complexity

Figure 1: Comparisons on different benchmarks and methods. (a) As existing datasets have not explored high-resolution images, particularly UHD images, our proposed 4K-Rain13k dataset will fill the gap in this research. (b) Model complexity and performance comparisons of the proposed method and other state-of-the-art models on the proposed 4K-Rain13k dataset in terms of PSNR, model parameters and FLOPs. The area of each circle denotes the number of model parameters. Since most approaches are unable to directly process UHD images, FLOP calculation is based on image sizes of $1024 \times 1024$. Our method achieves a better trade-off between efficiency and performance.

the harmony and consistency of the synthesized images, enabling them to better reflect the attributes of the original UHD image content. Based on this new dataset, we conduct extensive evaluations to analyze the performance of existing methods.

Furthermore, we find that when dealing with UHD images, most state-of-the-art (SOTA) methods often encounter memory overflow issues, making it difficult to perform full-resolution inference on consumer-grade GPUs. This motivates us to develop an effective and efficient method tailored for UHD image deraining. In this paper, we develop a simple yet effective architecture (called UDR-Mixer) to better solve this task, rather than relying on the self-attention mechanism that is computationally expensive in Transformers (Zamir et al., 2022). It is worth noting that our proposed method can achieve full-resolution inference of UHD images on consumer-grade GPUs.

The proposed UDR-Mixer consists of two parallel branches, each dedicated to exploring spatial and frequency representations to complement each other. In the main branch, we construct spatial feature mixing blocks (SFMB) as the core components, establishing global information perception through a simple yet effective feature rearrangement mechanism. Unlike strategies based on single-view spatial region rearrangement (Guo et al., 2022; Yu et al., 2022a), we recursively encode the entire image from different perspectives in multi-stage dimensional transformations and correlate multi-view features by permuting the tensor to better capture long-range pixel dependencies in UHD images. Simultaneously, we introduce an auxiliary branch composed of frequency feature mixing blocks (FFMB) to facilitate high-quality restoration of UHD images. Figure 1(b) illustrates that our method achieves favorable performance with a better trade-off between efficiency and performance.

The main contributions of this paper are summarized as follows:

- We propose the first high-quality UHD image deraining dataset (4K-Rain13k). Based on this dataset, we conduct a benchmark evaluation on existing methods for processing UHD images.

- We develop the spatial feature mixing block and the frequency feature mixing block to handle UHD images efficiently and formulate them into an end-to-end trainable network (UDR-Mixer) based on a dual-branch architecture for UHD image deraining.

- We quantitatively and qualitatively evaluate the proposed method on the proposed 4K-Rain13k dataset as well as real-world UHD images. Experimental results demonstrate that our approach achieves a favorable trade-off between performance and model complexity.

## 2 RELATED WORK

**Single image deraining**. When we revisit this field of single image deraining, numerous deraining approaches and benchmark datasets have been proposed in recent years with demonstrated success (Chen et al., 2023c). Several classic benchmark datasets are widely adopted to evaluate single image deraining performance, such as Rain200L/H (Yang et al., 2017), DID-Data (Zhang & Patel, 2018), DDN-Data (Fu et al., 2017) and Rain13k (Jiang et al., 2020). These early benchmark datasets consist of lower resolution images (1K or less). However, in this field, there is a lack of exploration specifically for higher resolution images, particularly UHD images. Furthermore, when dealing with UHD images, existing SOTA methods frequently encounter memory overflow issues, preventing them from conducting full-resolution inference on consumer-grade GPUs.

**UHD image processing**. With the development of photography equipment, UHD image processing has emerged as a new research trend in recent years (Zheng et al., 2021; Yu et al., 2022b; Wu et al., 2024). Zheng *et al.* (Zheng et al., 2021) formulated the UHD image dehazing network using multi-guide bilateral learning. Zhang *et al.* (Zhang et al., 2021) explored the task of image super-resolution for UHD resolutions, and further created two large-scale image datasets, UHDSR4K and UHDSR8K. Ren *et al.* (Ren et al., 2023) developed a multi-scale separable network to address UHD deblurring problem. The UHD low-light image enhancement task has also received increasing attention from researchers, and representative datasets include UHD-LOL (Wang et al., 2023) and UHD-LL (Li et al., 2023). Beyond that, other related tasks have focused on the application of UHD images, e.g., reflection removal and HDR reconstruction (Zheng et al., 2021). To the best of our knowledge, we first focus on the task of removing rain from UHD images, and we propose both a benchmark dataset and a baseline method.

**Vision MLP**. Given the high computational cost of self-attention mechanism in vision Transformers (ViT), several researchers have designed efficient vision models comprising solely multi-layer perceptrons (MLPs). For example, MLP-Mixer (Tolstikhin et al., 2021) utilizes a straightforward token-mixing MLP instead of self-attention in ViT, leading to an all-MLP network. It employs token-mixing MLP to capture token relationships and channel-mixing MLP to capture channel relationships. Afterwards, some studies further improve the performance of MLP-based models by designing other architectures, such as gMLP (Liu et al., 2021a) and Hire-MLP (Guo et al., 2022). Recently, Tu *et al.* (Tu et al., 2022) formulated a multi-axis MLP-based framework MAXIM for image processing tasks. Wu *et al.* (Wu et al., 2024) developed an efficient MixNet for UHD low-light image enhancement by modeling global and local feature dependencies. Inspired by these works, we leverage the vision MLP-like architecture to flexibly handle UHD image deraining.

## 3 UHD IMAGE DERAINING DATASET CONSTRUCTION

To evaluate the performance of existing approaches on the UHD image deraining problem, we first create a large-scale benchmark dataset named 4K-Rain13k. We note that existing low-resolution rain datasets (Yang et al., 2017; Fu et al., 2017; Zhang & Patel, 2018; Zhang et al., 2019; Jiang et al., 2020; Liu et al., 2021b) simply add rain streaks into the clear backgrounds to obtain rainy images. However, this copy-and-pasting approach is not suitable for synthesizing UHD rainy images due to the *geometric inconsistency* between the low-resolution and high-resolution image synthesis processes. Thus, we develop an effective method for synthesizing rainy images tailored for UHD images, aiming to achieve more realistic visual effects. Our method involves background collection, rain streak generation and geometric transformation, which will be presented below.

**Background collection**. We collect numerous clear backgrounds using a Python program based on Scrapy to download high-resolution images from the web and various devices. Our ground-truths includes a wide range of typical daytime and nighttime scenes in urban locations (e.g., buildings, streets, cityscapes) as well as natural landscapes (e.g., lakes, hills, and vegetations).

**Rain streak generation**. The diversity and fidelity of rain play crucial roles in the synthesis of rain streaks. Instead of using Photoshop software to render rain streaks, we synthesize corresponding rainy images by modeling the generation of rain streaks as a motion blur process to ensure diversity. In addition, we take into account the transparency of rain layer to ensure fidelity through alpha blending.

**Geometric transformation**. In fact, there is an easily overlooked problem of geometric inconsistency in the synthesis of low resolution and high-resolution rainy images, with noticeable discrepancies in

Table 1: Comparison between existing image deraining datasets and our proposed 4K-Rain13k dataset. 'Number': the number of paired images. 'Resolution': the average resolution of the dataset.

| Dataset | Year | Number | Avg. Resolution |
|---|---|---|---|
| Rain200L/H (Yang et al., 2017) | 2017 | 2.0K | $435 \times 366$ |
| DDN-Data (Fu et al., 2017) | 2017 | 13.0K | $489 \times 428$ |
| DID-Data (Zhang & Patel, 2018) | 2018 | 13.2K | $512 \times 512$ |
| Rain800 (Zhang et al., 2019) | 2019 | 0.8K | $518 \times 419$ |
| SPA-Data (Wang et al., 2019) | 2019 | 29.5K | $256 \times 256$ |
| Rain13k (Jiang et al., 2020) | 2020 | 13.7K | $482 \times 419$ |
| RainDirection (Liu et al., 2021b) | 2021 | 3.3K | $1945 \times 1444$ |
| RainDS (Quan et al., 2021) | 2021 | 1.4K | $818 \times 460$ |
| GT-RAIN (Ba et al., 2022) | 2022 | 31.5K | $666 \times 339$ |
| LHP-Rain (Guo et al., 2023) | 2023 | 1.0M | $1920 \times 1080$ |
| HQ-RAIN (Chen et al., 2023c) | 2023 | 5.0K | $1367 \times 931$ |
| 4K-Rain13k (Ours) | 2024 | 13.0K | $3840 \times 2160$ |

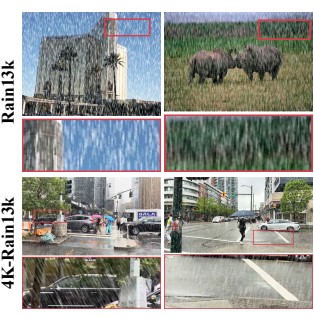

Figure 2: Sample images from the Rain13k dataset (Jiang et al., 2020) and our 4K-Rain13k dataset.

the length and thickness of rain streaks. Specifically, in high-resolution 4K images, with more pixel information available, rain streaks tend to appear longer and slighter as each rain-effected region can be accurately represented. In low-resolution images with fewer pixels, rain streaks of the same length may appear shorter or thicker due to blurring, resulting in a rougher and more ambiguous appearance.

To this end, we introduce geometric transformation operations to adjust the scale of rain streaks on the UHD images. By applying simple geometric transformations such as scaling, we aim to harmonize the proportions and sizes of rain streaks in the synthesized images with those observed in the high-resolution 4K images. This step helps alleviate the geometric disparities caused by varying image resolutions, ensuring that the rain streak patterns maintain their intended appearance and spatial relationships during the image synthesis process. We present sample images in Figure 2.

**Benchmark statistics**. Our proposed 4K-Rain13k dataset contains 12,500 synthetic training pairs and 500 test images at 4K resolution ($3840 \times 2160$). The training and test partitions are distinct in terms of their scenes and data, with no overlap. Table 1 presents a comparison between our dataset and existing public datasets. Following (Chen et al., 2023c), we utilize the Kullback-Leibler divergence (KLD), also known as relative entropy, to quantify the difference between the distribution of synthetic images and real images. Due to limited space, we provide the analysis results in the Appendix A.3. The results show that our 4K-Rain13k dataset is closer to the distribution of real-world rainy scenes.

## 4 PROPOSED UDR-MIXER

In this section, we develop an effective and efficient method (called UDR-Mixer) for UHD image deraining. We first describe the overall pipeline, and then present the details of two main components, i.e., spatial feature mixing blocks (SFMB) and frequency feature mixing blocks (FFMB).

### 4.1 OVERALL PIPELINE

As shown in Figure 3, UDR-Mixer is an end-to-end dual-branch parallel network architecture that models UHD images by exploring both spatial and frequency domain information. Specifically, we first embed an input rainy image $I_{rain} \in \mathbb{R}^{H \times W \times 3}$ into the feature space $F_0 \in \mathbb{R}^{H \times W \times C}$ through a $3 \times 3$ convolution layer, where $H$, $W$ and $C$ represents the height, width, and channel, respectively. To reduce computational complexity in high-resolution images, following previous studies (Li et al., 2023; Wu et al., 2024), we employ a PixelUnshuffle operation to downsample the features to $1/4$ of the original resolution. Then, the low-level features are processed by an encoder-decoder network consisting of $N_i$ SFMBs with a $2\times$ downsample operation and a $2\times$ upsample operation to produce output features.

To alleviate the issue of losing image details caused by straightforward downsampling operations, we further introduce an auxiliary branch to help UHD image reconstruction. Specifically, we stack $N_i$ FFMBs to excavate the frequency information of the full-resolution UHD image. Then, the learned deep features are fed to the decoder network of the main branch for guiding latent clear image restoration. Finally, the output features are obtained to estimate the derained image using a $3 \times 3$ convolutional layer followed by a PixelShuffle operation. To supervise the training process, we

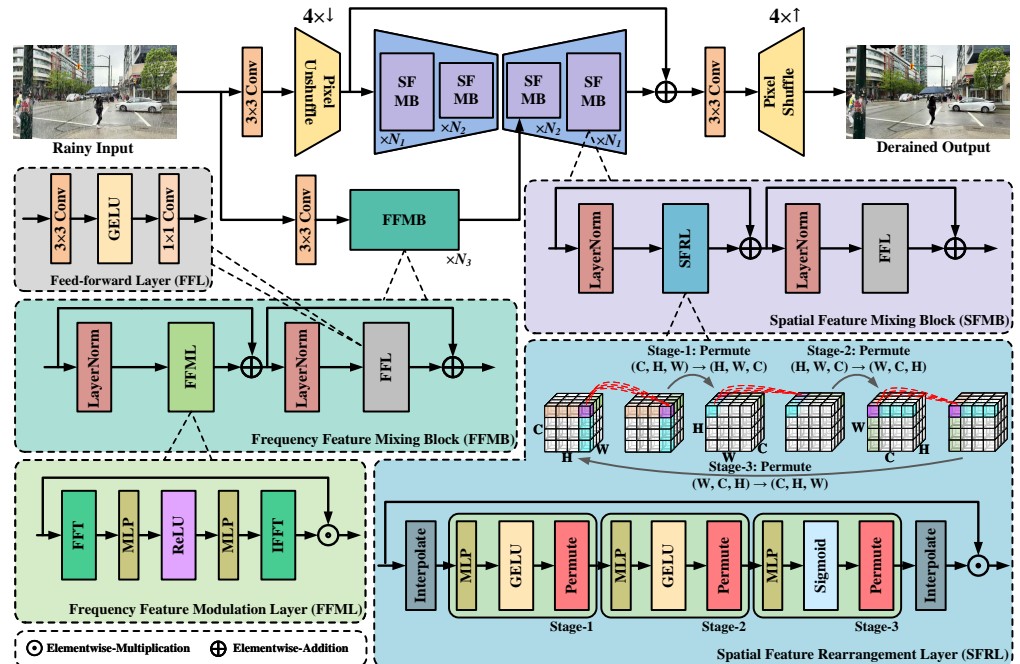

Figure 3: The overall architecture of the proposed UDR-Mixer for UHD image deraining, which mainly contains pixel unshuffle/shuffle operations and MLP-based components, i.e., spatial feature mixing blocks (SFMB) and frequency feature mixing blocks (FFMB).

employ the $L_1$ loss as the objective function:

$$\mathcal{L} = \|I_{derain} - I_{gt}\|_1 , \tag{1}$$

where $I_{gt}$ denotes the ground-truth image, and $\| \cdot \|_1$ denotes the $L_1$-norm.

## 4.2 SPATIAL FEATURE MIXING BLOCK

Inspired by MLP-Mixer (Tolstikhin et al., 2021), we introduce a new backbone module to encode feature information. Firstly, we develop a simple yet effective SFMB to aggregate global spatial information. Given the input feature $\mathbf{X}_{l-1} \in \mathbb{R}^{H \times W \times C}$, the proposed SFMB can be formulated as:

$$\begin{aligned}
\mathbf{X}'_l &= \mathbf{X}_{l-1} + \text{SFRL}\left(\text{LN}\left(\mathbf{X}_{l-1}\right)\right), \\
\mathbf{X}_l &= \mathbf{X}'_l + \text{FFL}\left(\text{LN}\left(\mathbf{X}'_l\right)\right),
\end{aligned} \tag{2}$$

where LN refers to the layer normalization. $\mathbf{X}'_l$ and $\mathbf{X}_l$ are output feature from the spatial feature rearrangement layer (SFRL) and feed-forward layer (FFL).

**Spatial feature rearrangement layer**. To model global spatial information with lower computational costs, we develop a SFRL based on continuous rearrangements and channel-mixing MLPs. Different from Hire-MLP (Guo et al., 2022) that utilizes region rearrangement strategy, we introduce a more flexible mechanism via dimension transformation operations, which directly considers the spatial properties between pixels in 3D feature maps. This enables the model to progressively capture global features across the entire image by scrolling, thereby establishing a gradual perception of long-range information from UHD images (see Figure 3). Specifically, we first normalize the input features, and then perform multi-stage dimension transformations to rotate the spatial perspective of the tensor across three dimensions of $H$, $W$ and $C$. Here, the 3D feature map undergoes recursive encoding from $(C, H, W)$ to $(H, W, C)$ and then to $(W, C, H)$, enabling the capture of global spatial information through multi-view dimensions. Finally, we adjust the feature map to the original resolution, and interact with the input features to activate useful features. Mathematically, given an input feature $\mathbf{F}_0$, the feature propagation process of SFRL can be expressed as:

$$\begin{aligned}
\mathbf{F}_l &= \text{Interpolate}\left(\mathbf{F}_0\right), \\
\mathbf{F}'_l &= \left[\mathcal{P}\left(\text{GELU}\left(\text{MLP}\left(\mathbf{F}_l\right)\right)\right)\right]_{\times 2} + \mathcal{P}\left(\text{Sigmoid}\left(\text{MLP}\left(\mathbf{F}_l\right)\right)\right), \\
\hat{\mathbf{F}}_l &= \text{Interpolate}\left(\mathbf{F}'_l\right) \odot \mathbf{F}_0,
\end{aligned} \tag{3}$$

where Interpolate$(\cdot)$ and $\mathcal{P}(\cdot)$ denote the interpolation and permute functions. GELU$(\cdot)$ and Sigmoid$(\cdot)$ represents GELU and Sigmoid functions. MLP is a $1 \times 1$ convolution layer. $\odot$ represents element-wise multiplication.

**Feed-forward layer**. Similar to vision Transformer (ViT) (Sharir et al., 2021), FFL performs dimension reduction and non-linear transformations. Here, we adopt a FFL to transform features into compact representations, which is defined as follows:

$$\mathbf{F}_l = \text{Conv}_{1 \times 1} \left( \text{GELU} \left( \text{Conv}_{3 \times 3} \left( (\mathbf{F}_0) \right) \right) \right), \tag{4}$$

where $\text{Conv}_{3 \times 3}$ is a $3 \times 3$ convolution layer. The initial $3 \times 3$ convolution is used to enhance locality and increase the number of channels for channel mixing. The later $1 \times 1$ convolution is adopted for reducing the channels back to the original input dimension.

## 4.3 Frequency feature mixing block

We note that existing methods (Wang et al., 2023; Li et al., 2023; Ren et al., 2023; Wu et al., 2024) mostly employ direct downsampling of UHD images to create low-resolution versions, aiming to reduce computational burden. In such cases, the full-resolution restoration process is predominantly governed by information learned solely from the low-resolution images, resulting in suboptimal performance and a tendency to lose high-frequency details, which are abundant in UHD images. To this end, we develop the FFMB as a unit for the auxiliary branch. It leverages the frequency domain information of the full-resolution images and guides the decoding restoration process of the main branch composed of SFMB. Given the input feature $\mathbf{Y}_{l-1} \in \mathbb{R}^{H \times \tilde{W} \times C}$, the proposed FFMB can be formulated as:

$$\begin{aligned} \mathbf{Y}_l' &= \mathbf{Y}_{l-1} + \text{FFML} \left( \text{LN} \left( \mathbf{Y}_{l-1} \right) \right), \\ \mathbf{Y}_l &= \mathbf{Y}_l' + \text{FFL} \left( \text{LN} \left( \mathbf{Y}_l' \right) \right), \end{aligned} \tag{5}$$

where $\mathbf{Y}_l'$ and $\mathbf{Y}_l$ are output feature from the frequency feature modulation layer (FFML) and FFL. The structure of FFL remains the same as that in SFMB.

**Frequency feature modulation layer**. According to the convolution theorem, convolution in one domain is mathematically equivalent to the Hadamard product in its corresponding Fourier domain (Huang et al., 2023). This also motivates us to introduce a FFML for implementing frequency-space manipulation. Given an input feature $\mathbf{F}_0$, we employ Fast Fourier Transform (FFT) to obtain the corresponding frequency representations. Then, we adopt two stacks of MLP layers with a ReLU layer in between. Finally, we perform an inverse FFT and interact with the input features to obtain updated feature representations in the original latent space. This process can be formulated as:

$$\begin{aligned} \mathbf{F}_l &= \mathcal{F}^{-1} \left( \text{MLP} \left( \text{ReLU} \left( \text{MLP} \left( \mathcal{F} \left( \mathbf{F}_0 \right) \right) \right) \right) \right), \\ \hat{\mathbf{F}}_l &= \mathbf{F}_l \odot \mathbf{F}_0, \end{aligned} \tag{6}$$

where $\mathcal{F}(\cdot)$ denotes the FFT and $\mathcal{F}^{-1}(\cdot)$ denotes the inverse FFT.

# 5 Experiments

In this section, we first present the experimental settings of our proposed UDR-Mixer. Then we conduct a benchmark study on our method and other comparative methods. More results can be found in the supplementary material. The code and dataset will be available to the public.

## 5.1 Experimental settings

**Implementation details**. In our model, $\{N_1, N_2, N_3\}$ are set to $\{2, 2, 4\}$. The initial number of feature channels for the main and auxiliary branches is set to 48 and 64, respectively. We conduct model training on four NVIDIA GeForce RTX 3090 GPUs with 24GB memory. In total, we perform 500 epochs of training. During the training, we adopt the Adam optimizer with a learning rate of $2 \times 10^{-4}$. The patch size is set to be $768 \times 768$ pixels and the batch size is set to be 8. To augment the training data, we apply random horizontal and vertical flips. For testing UHD images, we use one NVIDIA TESLA V100 with 32GB memory.

Table 2: Quantitative evaluations on the proposed 4K-Rain13k dataset. "Params" and "FLOPs" represent the number of trainable model parameters (in M) and FLOPs (in G), respectively. The results of FLOPs are tested on the images with $1024 \times 1024$ pixels.

| Methods | | Venue | 4K-Rain13k | | | Complexity | |
|---|---|---|---|---|---|---|---|
| | | | PSNR ↑ | SSIM ↑ | MSE ↓ | Params | FLOPs |
| Rainy input | | | 21.14 | 0.7594 | 812.98 | - | - |
| Prior-based methods | DSC (Luo et al., 2015) | ICCV'15 | 22.93 | 0.6299 | 498.63 | - | - |
| CNN-based methods | LPNet (Fu et al., 2019) | TNNLS'19 | 27.86 | 0.8924 | 171.33 | 0.03 | 57.1 |
| | JORDER-E (Yang et al., 2019) | TPAMI'19 | 31.01 | 0.9119 | 104.30 | 4.21 | 4363.3 |
| | RCDNet (Wang et al., 2020) | CVPR'20 | 31.41 | 0.9215 | 95.01 | 3.17 | 3400.3 |
| | SPDNet (Yi et al., 2021) | ICCV'21 | 32.38 | 0.9233 | 77.49 | 3.04 | 1428.8 |
| Transformer-based methods | IDT (Xiao et al., 2022) | TPAMI'22 | 32.91 | 0.9479 | 57.04 | 16.41 | - |
| | Restormer (Zamir et al., 2022) | CVPR'22 | 33.70 | 0.9344 | 58.98 | 26.12 | 2478.1 |
| | DRSformer (Chen et al., 2023b) | CVPR'23 | 33.47 | 0.9329 | 62.91 | 33.65 | 3887.8 |
| | UDR-S2Former (Chen et al., 2023a) | ICCV'23 | 33.36 | 0.9458 | 50.69 | 8.53 | 395.8 |
| Ours | UDR-Mixer | - | **34.30** | **0.9505** | **42.03** | 4.90 | 200.1 |

(a) Rainy input    (b) JORDER-E    (c) RCDNet    (d) SPDNet    (e) IDT

(f) Restormer    (g) DRSformer    (h) UDR-S2Former    (i) Ours    (j) Ground truth

Figure 4: Visual quality comparison on the 4K-Rain13k dataset. Compared with the derained results in (b)-(h), our method recovers a high-quality image with clearer details.

**Compared methods**. We compare our approach with prior-based algorithms (i.e., DSC (Luo et al., 2015)), CNN-based networks (i.e., LPNet (Fu et al., 2019), JORDER-E (Yang et al., 2019), RCDNet (Wang et al., 2020), SPDNet (Yi et al., 2021)), and Transformer-based models (i.e., IDT (Xiao et al., 2022), Restormer (Zamir et al., 2022) , DRSformer (Chen et al., 2023b), and UDR-S2Former (Chen et al., 2023a)). For fair comparison, we utilize the official released code of these approaches. All deep learning-based methods are retrained on the proposed 4K-Rain13k dataset with their default settings for an equal number of epochs. We uniformly select the weights from their final training epoch for testing purposes. Note that for some approaches (JORDER-E (Yang et al., 2019), RCDNet (Wang et al., 2020), SPDNet (Yi et al., 2021), Restormer (Zamir et al., 2022), and DRSformer (Chen et al., 2023b)), we are unable to infer full-resolution results on UHD images. Following previous UHD studies (Zheng et al., 2021; Li et al., 2023), we adopt a splitting-and-stitching strategy, which involves splitting the input into multiple patches and then stitching the results together.

**Evaluation metrics**. For the 4K-Rain13k benchmark with ground truth images, we employ full-reference metrics PSNR (Huynh-Thu & Ghanbari, 2008), SSIM (Wang et al., 2004) and MSE to evaluate the image quality of each restored results. For the real-world scenes without ground truth images, we adopt the non-reference metrics NIQE (Mittal et al., 2012a), PIQE (Venkatanath et al., 2015), and BRISQUE (Mittal et al., 2012b). Higher PSNR and SSIM values signify better restoration quality, while lower MSE, NIQE, PIQE and BRISQUE scores indicate better perceptual quality. We also test the trainable parameters and FLOPs to analyze the computational complexity of the model.

## 5.2 COMPARISONS WITH THE STATE OF THE ART

**Evaluations on the proposed 4K-Rain13k**. Table 2 presents the quantitative results of different approaches on the proposed 4K-Rain13k dataset. It can be observed that our proposed UDR-Mixer achieves the highest PSNR and SSIM values while maintaining the lowest MSE value, indicating the superiority of our method in rain removal from UHD images. Specifically, our method outperforms the state-of-the-art UDR-S2Former (Chen et al., 2023a) by 0.94dB in terms of PSNR, while utilizing

Table 3: Quantitative evaluations on real rainy images. For all methods, we use the pre-trained models from the 4K-Rain13k dataset to evaluate the image deraining capabilities in real-world scenarios.

| Methods | Input | RCDNet | SPDNet | IDT | Restormer | DRSformer | UDR-S2Former | Ours |
|---|---|---|---|---|---|---|---|---|
| NIQE ↓ | 8.208 | 9.997 | 9.917 | 9.067 | 8.636 | 8.493 | 8.104 | **7.509** |
| PIQE ↓ | 54.863 | 63.816 | 64.774 | 55.049 | 60.335 | 60.441 | 55.204 | **53.104** |
| BRISQUE ↓ | 67.855 | 71.967 | 67.461 | 67.100 | 65.102 | 63.823 | 65.177 | **53.192** |

Table 4: Quantitative evaluations on the RainDS-RS dataset, which contains Syn-RS and Real-RS.

| Methods | JORDER-E | MSPFN | MPRNet | Uformer | Restormer | IDT | UDR-S2Former | Ours |
|---|---|---|---|---|---|---|---|---|
| PSNR ↑ | 30.11 | 32.53 | 34.05 | 33.79 | 34.41 | 34.56 | 35.15 | **35.72** |
| SSIM ↑ | 0.819 | 0.851 | 0.859 | 0.851 | 0.861 | 0.863 | 0.867 | **0.868** |

(a) Rainy input    (b) SPDNet    (c) Restormer    (d) DRSformer    (e) UDR-S2Former    (f) Ours

Figure 5: Visual quality comparison on a real-world UHD rainy image from the collected 4K-RealRain. Compared with the derained results in (b)-(e), our method recovers a clearer image.

fewer network parameters and lower FLOPs. This advantage is even more pronounced compared to other heavy models such as Restormer (Zamir et al., 2022) and DRSformer (Chen et al., 2023b). In other words, our proposed method achieves a better trade-off between restoration performance and model efficiency. In Figure 4, we further present the visual results of different methods. We observe that CNN-based methods struggle to recover texture details, such as those in building areas, under the influence of densely packed rain streaks. Additionally, despite its ability to model global information, UDR-S2Former (Chen et al., 2023a), as a Transformer-based method, still exhibits sensitivity to spatially-long rain streaks present in UHD images, resulting in residual rain artifacts. In contrast, our approach produces clearer images while preserving high-frequency information. This is attributed to the complementary advantages of dual domain branches in our UDR-Mixer.

**Evaluations on real 4K rainy images**. To further evaluate the generalization capability of various deraining methods in real rainy scenes, we collect 320 real 4K rainy images from the Internet and real-world sources, referred to as 4K-RealRain. These scenes mostly originate from high-definition captures using smartphones. The quantitative results for different methods are reported in Table 3. Clearly, our method achieves the lowest values across three metrics: NIQE, PIQE, and BRISQUE. This indicates that, compared to other models, the output results from our UDR-Mixer exhibit clearer content and better perceptual quality in real rainy scenes. Figure 5 displays a comparison of visual results. Our method effectively removes most rain streaks and exhibits visually pleasing restoration effects, indicating its capability to generalize well to unseen real-world data types.

**Evaluations on low-resolution benchmarks**. We further validate the scalability of our method on low-resolution benchmarks. Following (Chen et al., 2023a), we conduct experiments on the RainDS-RS dataset (Quan et al., 2021), which contains Syn-RS and Real-RS subnets. Here, we adjust our UDR-Mixer model for a fair comparison. Specifically, we remove the pixel unshuffle/shuffle

Table 5: Ablation comparison on different variants of our UDR-Mixer on the 4K-Rain13k dataset.

| Methods | SFMB | | | | | | FFMB | PSNR | SSIM |
|---|---|---|---|---|---|---|---|---|---|
| | Spatial Shift | H-Region Rearrange | W-Region Rearrange | Dimension Rearrange | | | | | |
| | | | | Stage-1 | Stage-2 | Stage-3 | | | |
| (i) | | | | | | | | 33.41 | 0.9417 |
| (ii) | ✔ | | | | | | | 33.65 | 0.9433 |
| (iii) | | ✔ | | | | | | 34.09 | 0.9437 |
| (iv) | | | ✔ | | | | | 33.74 | 0.9413 |
| (v) | | | | ✔ | | | | 32.41 | 0.9394 |
| (vi) | | | | ✔ | ✔ | | | 32.50 | 0.9409 |
| (vii) | | | | ✔ | ✔ | ✔ | | 34.15 | 0.9465 |
| (viii, Ours) | | | | ✔ | ✔ | ✔ | ✔ | **34.30** | **0.9505** |

operations used for UHD images in the main branch, while keeping other components consistent. For distinction, we name it UDR-Mixer-L. According to the quantitative results in Table 4, our proposed method not only demonstrates satisfactory deraining effects on 4K images but also proves effective in low-resolution scenes.

We also validate the generalization performance of different methods on an existing real low-resolution rainy dataset, RE-RAIN (Chen et al., 2023c). Visual results presented in Figure 6 indicate that the restoration results of most methods still exhibit varying degrees of rain residue. We observe that while most Transformer-based methods exhibit competitive performance on synthetic datasets, their deraining ability significantly decreases in real-world scenes. In contrast, our UDR-Mixer not only demonstrates satisfactory deraining effects on 4K images but also proves effective in real low-resolution scenes. Due to the flexibility of the proposed SFRL, our proposed method can be better applied to real rain scenarios across diverse image resolutions.

### 5.3 ABLATION STUDY

We conduct ablation studies to examine the effect of our method, training all variant models under same settings on the 4K-Rain13k dataset to ensure fairness.

**Effectiveness of rearrange strategy in SFMB**. We first replace the proposed SFMBs with residual blocks that have comparable parameters as the baseline model (i). Table 5 shows that our method improves restoration performance better compared to the baseline model by introducing SFMB. The feature rearrangement strategy is a critical component of our proposed SFMB. Here, we compare with recent MLP-based feature rearrangement methods, including spatial shift (Yu et al., 2022a), height-direction region rearrange and width-direction region rearrange (Guo et al., 2022). Compared to methods (ii-iv), our dimension rearrangement approach yields superior quantitative results. The visual results in Figure 7 (b-d) and (f) also demonstrate that our method not only effectively removes complex rain streaks but also better preserves the fine details of the image. The reason behind this lies in our method implicitly enhancing the capture of multi-view features through dimension transformation, making it more suitable for modeling long-range spatial relationships in UHD images.

**Effect of the number of permute stages**. We further analyze the effect of the number of permute stages in the SFMB. Note that we utilize Permute operations to rotate rotate 3D feature maps between adjacent stages. When stage=1, the model (v) can only capture the single-view features. We find that through multiple stages of recursive encoding, features learned from three perspectives are effectively correlated, thus aiding in further boosting the image restoration performance.

**Effectiveness of FFMB**. The FFMB in the auxiliary branch is used to better explore frequency information in our UDR-Mixer for high-quality UHD image restoration. To demonstrate the effectiveness of this branch, we remove this component and investigate its influence in Table 5. In comparison to our approach, the restoration performance of model (vii) is suboptimal. In addition, Figures 7 (e) and (f) also show that our method generates much clearer details.

### 5.4 DISCUSSIONS WITH THE CLOSELY-RELATED METHOD

We note that the recent method MAXIM (Tu et al., 2022) proposes a multi-axis MLP based architecture to solve imgae deraining. Different from MAXIM that employs the multi-axis gated strategy, our

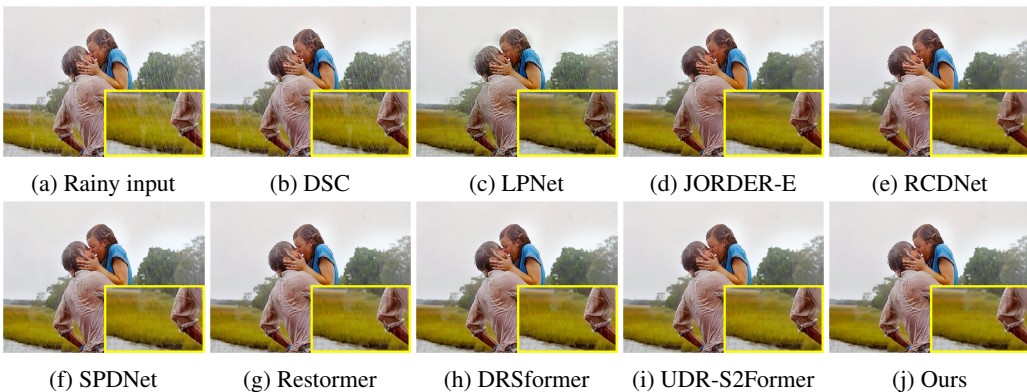

Figure 6: Visual quality comparison on a real low-resolution rainy image from the RE-RAIN dataset. Compared with the derained results in (b)-(i), our method recovers a clearer image. Zooming in the figures offers a better view at the deraining capability.

Table 6: Comparison of generalization results between our method and MAXIM (Tu et al., 2022) on real rainy images.

| Methods | NIQE↓ | PIQE↓ | BRISQUE↓ | Params | FLOPs |
|---|---|---|---|---|---|
| MAXIM | 4.564 | 12.567 | 23.929 | 14.1M | 216G |
| UDR-Mixer-L | **4.384** | **7.160** | **21.885** | 2.6M | 65G |

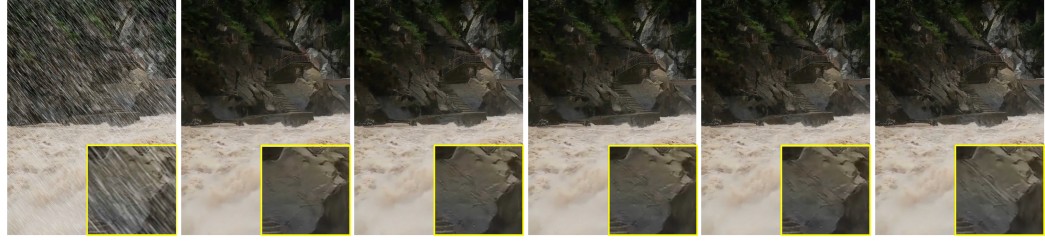

(a) Rainy input   (b) Spatial shift   (c) H-region RR   (d) W-region RR   (e) w/o FFMB   (f) Ours

Figure 7: Visual comparison on the rearrange strategy in SFMB (b-d) and the proposed FFMB (e).

method utilizes a simple yet effective dimension rearrange mechanism to capture spatial information. First, we report the model complexity in Table 6. Compared to MAXIM, our UDR-Mixer-L achieves a 81.6% reduction in model parameters while decreasing FLOPs by 69.9%. Note that as the training code of MAXIM is not available, we do not benchmark this approach on our proposed 4K-Rain13k. Since the testing code of MAXIM and the pre-trained model on the Rain13k dataset (Jiang et al., 2020) are available, we compare their generalization ability of MAXIM and our method in real rainy scenes. As shown in Table 6, the rain removal results obtained by our method have better visual quality. More visual comparison results are provided in the Appendix A.11.

## 6 CONCLUDING REMARKS

This paper explores the task of UHD image deraining for the first time and proposes a high-quality dataset 4K-Rain13k to facilitate the performance comparison. Furthermore, we develop an efficient method UDR-Mixer for UHD image deraining. Our approach utilizes a dimension rearrange mechanism to establish the global spatial context of UHD images and combines it with the original frequency representation of UHD images to help image restoration. The benchmark results show that our model achieves a favorable trade-off between performance and model complexity.

**Limitations**. Although our method achieves favorable performance, it fails to handle the presence of fog-like rain accumulation in real rainy scenes. Future work will consider expanding 4K data with veiling effect and introducing physical models to guide enhancing the quality of image reconstruction.

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

# A APPENDIX

## A.1 MORE IMPLEMENTATION DETAILS ON DATASET CONSTRUCTION

**Rain Streak Generation**. We model the generation of rain streak layers as a motion blur process, leveraging two essential characteristics of rain streaks: their repeatability and directionality. Formally, this can be represented as:

$$\mathbf{S} = \mathbf{K}(l, \theta, w) * \mathbf{N}(n), \tag{7}$$

where $\mathbf{N}$ represents the rain mask derived from random noise $n$, where we utilize uniform random numbers combined with thresholds to adjust the noise level. The parameters $l$ and $\theta$ correspond to the length and angle of the motion blur kernel $\mathbf{K} \in \mathbb{R}^{p \times p}$. Additionally, a rotated diagonal kernel is applied with Gaussian blur to control the rain streak thickness $w$. The values for noise quantity $n$, rain length $l$, rain angle $\theta$, and rain thickness $w$ are sampled from the ranges $[100, 300]$, $[20, 40]$, $[40°, 120°]$, and $[3, 7]$, respectively. The symbol $*$ denotes the spatial convolution operator.

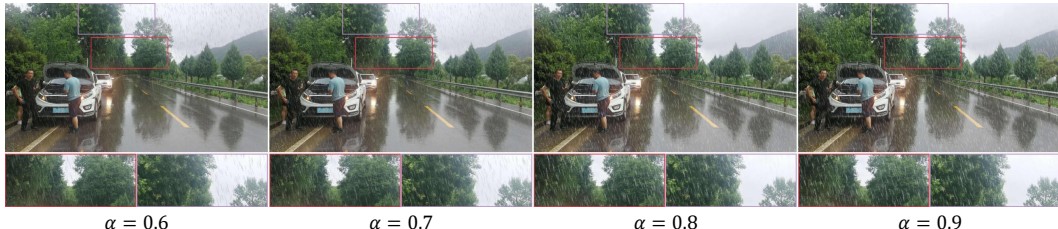

$\alpha = 0.6 \qquad\qquad \alpha = 0.7 \qquad\qquad \alpha = 0.8 \qquad\qquad \alpha = 0.9$

Figure 8: The effect of the blending ratio on the quality of the dataset.

**Geometric Transformation**. In the physical world, rain streaks are naturally fine and elongated, reflecting their slender geometry and high-speed movement. In high-resolution 4K images, with the increased pixel density and more detailed spatial representation, these characteristics can be captured more accurately. Each rain-affected region is represented with greater precision, allowing the streaks to appear longer and thinner, consistent with their real-world appearance. Conversely, in low-resolution images, the reduced pixel information limits the ability to preserve these fine details. Rain streaks that are physically the same length may appear shorter and thicker due to the loss of spatial resolution and the effects of blurring. This results in a coarser and more ambiguous depiction of rain streaks, where fine structures and elongated shapes are less distinguishable. This disparity highlights the importance of resolution in faithfully representing rain streaks and motivates the development of high-resolution synthesis approaches. Based on these real-world observations, we perform geometric transformation using scaling operations on the initial rain layer generated in the previous step.

**Alpha Blending**. We utilize alpha blending to combine the rain layer with the background layer. Here, the alpha value of each pixel in a layer determines the extent to which colors from underlying layers are visible through the current layer's color. Mathematically, this process is expressed as:

$$\mathbf{R}_{r,g,b} = \alpha \odot \mathbf{S} + (1 - \alpha) \odot \mathbf{B}, \qquad (8)$$

where $\alpha$ represents the blending ratio, which is empirically set within the range $[0.8, 0.9]$. The symbol $\odot$ indicates element-wise multiplication. The $R$, $G$, and $B$ channels are handled independently during the process. Figure 8 shows an example of synthesizing rainy images with different blending ratios. It is evident that when the blending ratios are set to 0.8 and 0.9, the rain streaks become invisible in the white sky region, resulting in greater harmony with real rainy images.

## A.2 MORE SAMPLE IMAGES

We further present several sample images from existing representative datasets (*i.e.*, Rain200L/H (Yang et al., 2017), DDN-Data (Fu et al., 2017), DID-Data (Zhang & Patel, 2018), Rain800 (Zhang et al., 2019), SPA-Data (Wang et al., 2019), Rain13k (Jiang et al., 2020), RainDirection (Liu et al., 2021b), RainDS (Quan et al., 2021), GT-RAIN (Ba et al., 2022), LHP-Rain (Guo et al., 2023)) and our proposed 4K-Rain13k dataset in Figure 9. Note that we maintain the original resolution ratio of different images. Our proposed new benchmark fills the gap in research on UHD image deraining.

## A.3 COMPARISONS WITH PREVIOUS DATASETS

We adopt the Kullback-Leibler Divergence (KLD) (Joyce, 2011), also known as relative entropy, to measure the difference between two probability distributions (*i.e.*, the synthetic image and real-world image). Figure 10 presents the comparison results of the representative synthetic benchmarks (Yang et al., 2017; Li et al., 2019; Zhang & Patel, 2018; Fu et al., 2017) and our benchmark, showing that our 4K-Rain13k is close to the distribution of real-world rainy images. The reason behind this is that 4K-Rain13k fully considers the harmony of the synthesized rainy images, thereby narrowing the domain gap between synthetic and real images.

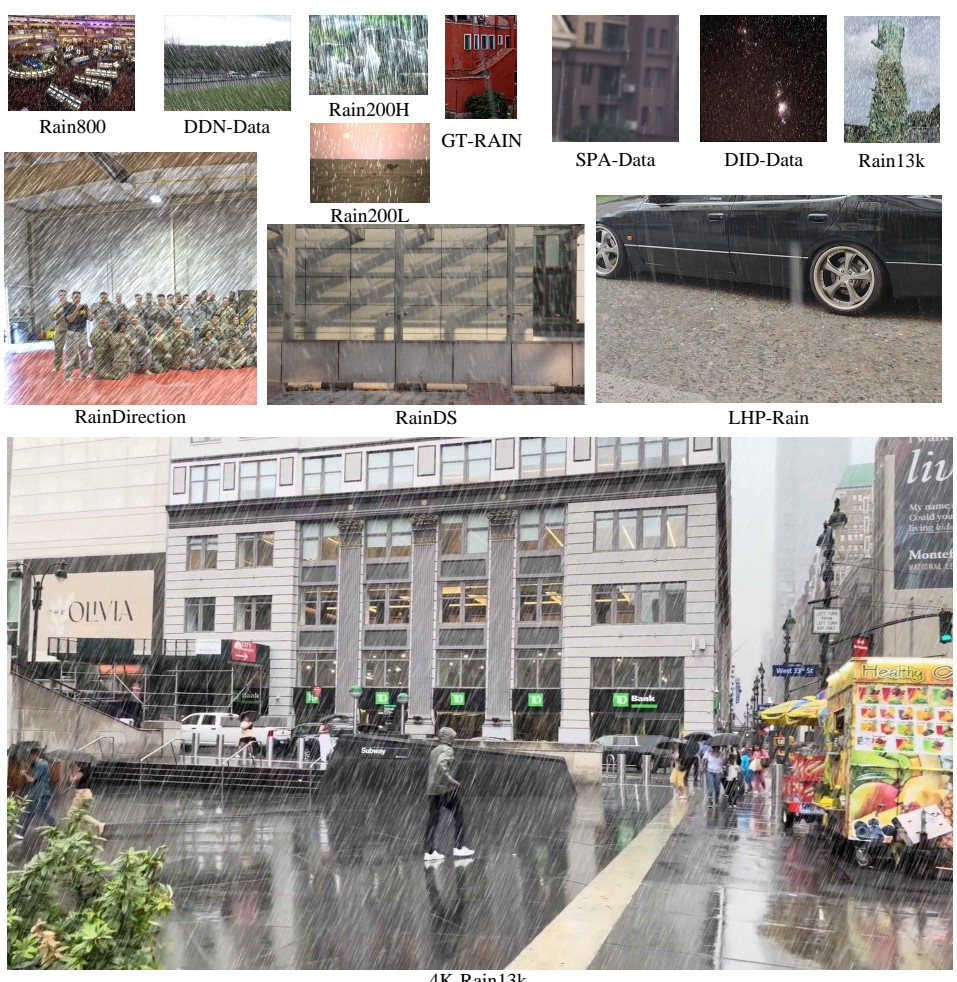

Figure 9: Example images from previous representative datasets(Yang et al., 2017; Fu et al., 2017; Zhang & Patel, 2018; Zhang et al., 2019; Wang et al., 2019; Jiang et al., 2020; Liu et al., 2021b; Ba et al., 2022; Guo et al., 2023; Quan et al., 2021) and our proposed 4K-Rain13k.

### A.4 GENERALIZATION ANALYSIS OF DATASETS

We compare the performance of our model trained on the proposed dataset and existing low-resolution dataset (Rain13k) and then test on real-world 4K rainy images. We have provided the comparison results in Figure 11. It can be observed that the model trained on our 4K-Rain13k performs better on real-world rainy images, indicating that our dataset effectively reduces the domain gap and enables the model to generalize better to real-world scenarios.

### A.5 ILLUSTRATION OF FEATURE REARRANGE STRATEGY

In the main paper, we compare with recent MLP-based feature rearrangement methods, including spatial shift (Yu et al., 2022a), height-direction region rearrange and width-direction region rearrange (Guo et al., 2022). Here, we present schematic diagrams of these three feature rearrangement strategies in Figure 12.

### A.6 DISCUSSIONS WITH THE CLOSELY-RELATED METHOD

We note that (Wu et al., 2024) develops a Global Feature Modulation Layer (GFML) for achieving dimension transformations. Compared to GFML, our SFRL has several differences: (1) **Flexible**

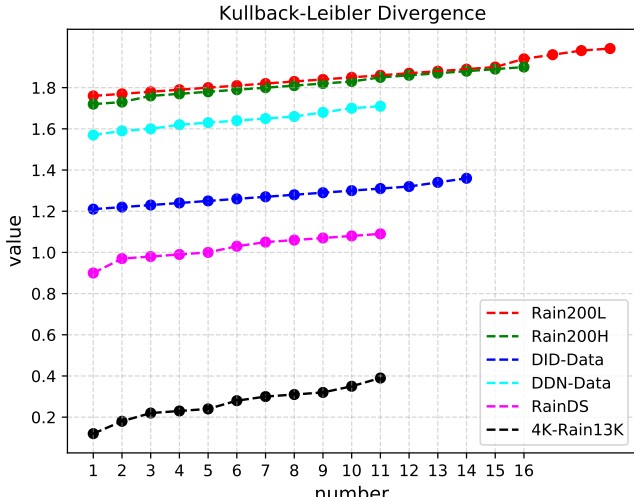

Figure 10: Comparison of Kullback CLeibler divergence (KLD) between different synthetic datasets and one real dataset. The vertical axis represents the value of KLD, and the horizontal axis represents the number of samples. Obviously, our proposed 4K-Rain13k obtains a lowest KLD score, indicating that our dataset has a smaller domain gap between the synthetic and real images compared to others.

**Rearrangement**: Unlike the fixed swapping of adjacent elements in GFML, SFRL rearranges the positions of all elements in the 3D feature map during each dimension transformation. This allows our method to flexibly capture global spatial information. (2) **Multi-Scale Integration**: While GFML operates at a single scale, SFRL is embedded within a multi-scale encoder-decoder network backbone, enabling more effective exploration of multi-scale information, which is crucial for image deraining. (3) **Enhanced UHD Restoration**: MixNet directly downsamples UHD images to learn spatial features, leading to detail loss. Our method addresses this issue by introducing the FFML, which enhances UHD image restoration quality through joint learning in both spatial and frequency information.

## A.7 EVALUATIONS ON THE LOW-RESOLUTION BENCHMARKS

We further evaluate our method on an existing low-resolution image deraining benchmark, RainDS (Quan et al., 2021). Table 8 presents the quantitative results of different approaches. Here, we refer to the results reported in (Chen et al., 2023a) (ICCV'23). It can be seen that our UDR-Mixer-L still achieves competitive performance while maintaining a lower model complexity.

## A.8 RUNTIME ANALYSIS

We calculate the runtime of our method and recent image deraining methods on 4K images. We conduct on a machine equipped with an NVIDIA RTX 4090 GPU. Tabale 7 shows that our method achieves lower inference time.

## A.9 APPLICATION-BASED EVALUATION

With the advancement of intelligent assisted driving systems, UHD images are increasingly being integrated into existing onboard devices. To investigate whether the image deraining process benefits downstream vision-based applications such as object detection, we apply mainstream object detection pre-trained models (YOLOv5) to evaluate the results.

Figure 13 shows that our approach not only reconstructs clear images but also enhances the accuracy of target recognition, such as the motorcycle category. Therefore, the method we developed is of significant importance for improving the practicality of real-world applications.

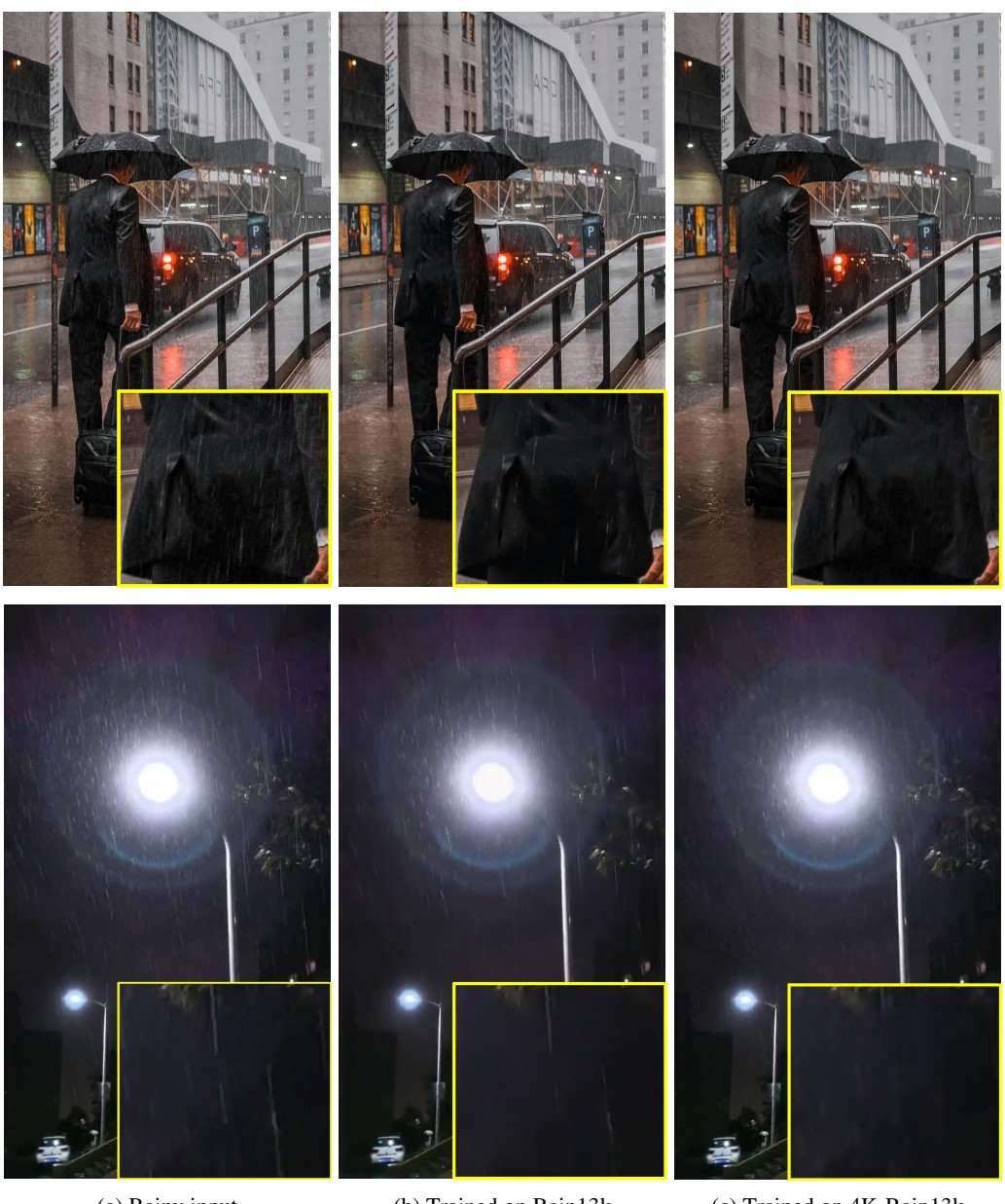

(a) Rainy input        (b) Trained on Rain13k        (c) Trained on 4K-Rain13k

Figure 11: Generalization results on real rainy images trained on the low-resolution Rain13k dataset and our proposed high-resolution 4K-Rain13k dataset.

### A.10 FAILURE CASE

As shown in Figure 14, our method is able to effectively remove rain streaks but fails to handle the presence of fog-like rain accumulation in real rainy scenes. In addition, the splashing effect of raind on the ground is also worth paying attention to in future work, which is crucial for downstream autonomous driving.

### A.11 MORE VISUAL COMPARISON RESULTS

In this section, we show more visual comparison results to demonstrate the effectiveness of the proposed method. Figure 15 shows the visual comparison results on real rainy images with MAXIM (Tu et al., 2022). It can be seen that our method successfully removes most rain streaks and generates

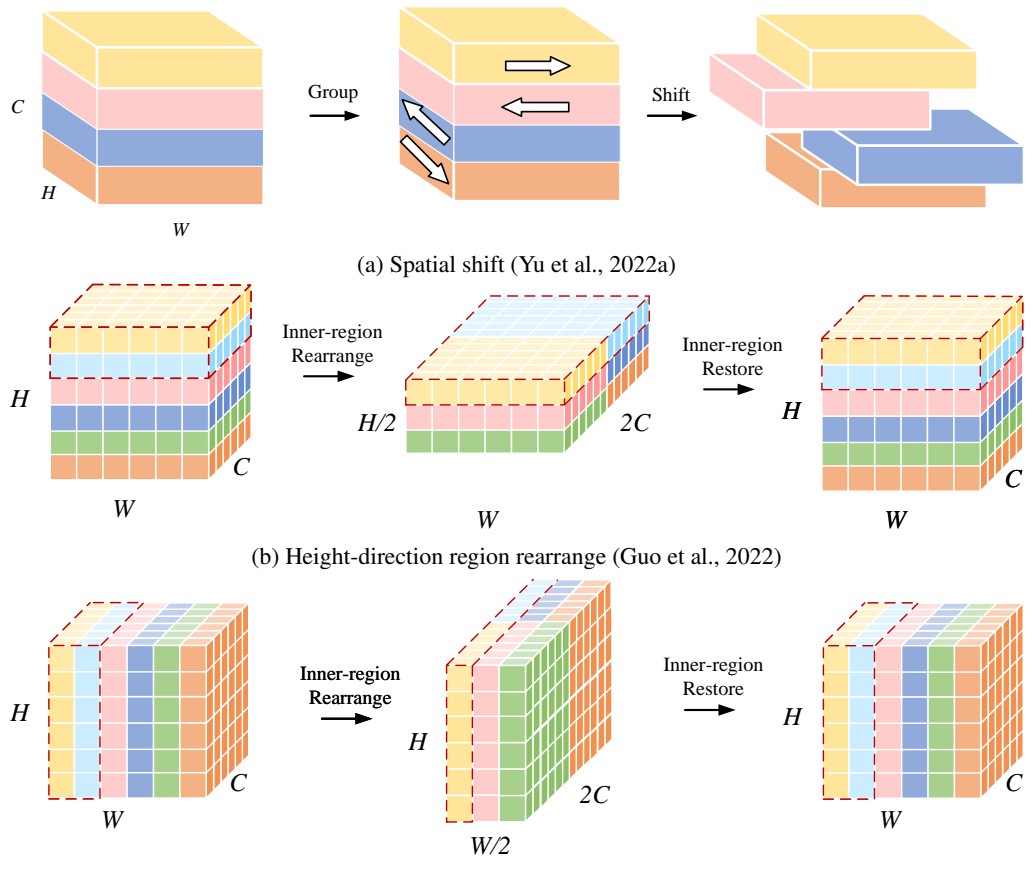

(a) Spatial shift (Yu et al., 2022a)

(b) Height-direction region rearrange (Guo et al., 2022)

(c) Width-direction region rearrange (Guo et al., 2022)

Figure 12: Illustration of other feature rearrange strategies.

Table 7: Comparison of running times of different models.

| Methods | IDT | Restormer | DRSformer | UDR-S2Former | UDR-Mixer(Ours) |
|---|---|---|---|---|---|
| Runtime(s) | 0.1677 | 0.6324 | 1.2061 | 0.8194 | 0.0072 |

a clearer image. Furthermore, Figures 16-17 show the visual comparison results on the proposed 4K-Rain13k dataset. Compared to other methods, our UDR-Mixer can generate high-quality deraining results with more accurate detail and texture recovery. Finally, Figures 18-19 show the visual comparison results on real rainy images. Our method can successfully remove complex and random rain streaks and own visual pleasant recovery results.

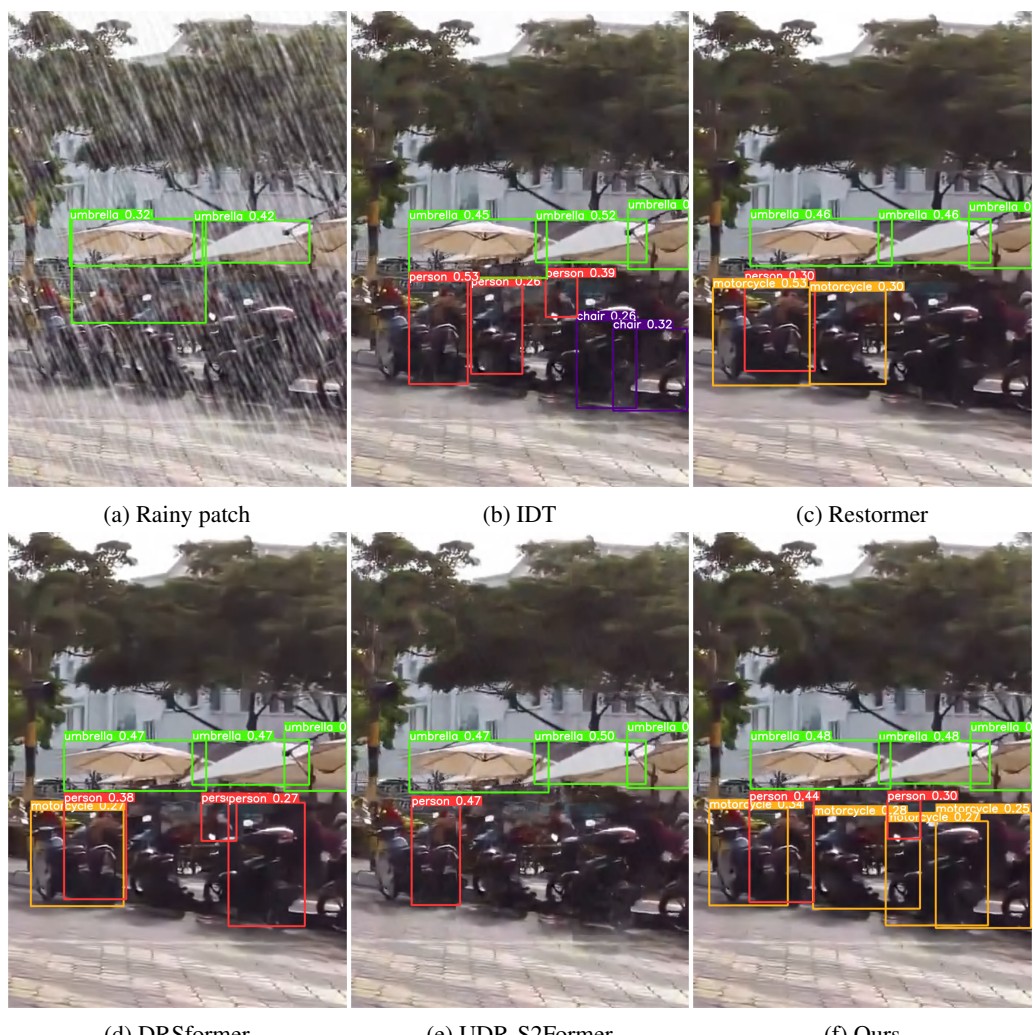

Figure 13: Object recognition results for the input rainy image and the derained images by different methods. Here, we crop a region from an UHD image for better comparison.

Table 8: Quantitative evaluations on the RainDS dataset. "Params" and "FLOPs" represent the number of trainable model parameters (in M) and FLOPs (in G), respectively. The results of FLOPs are tested on the images with $256 \times 256$ pixels.

| Method | Venue | Syn-RS | | Real-RS | | Average | | Params | FLOPs |
|---|---|---|---|---|---|---|---|---|---|
| | | PSNR | SSIM | PSNR | SSIM | PSNR | SSIM | | |
| GMM (Li et al., 2016) | CVPR'16 | 26.66 | 0.781 | 23.73 | 0.560 | 25.20 | 0.671 | - | - |
| JCAS (Gu et al., 2017) | ICCV'17 | 26.46 | 0.786 | 24.04 | 0.556 | 25.05 | 0.671 | - | - |
| DDN (Fu et al., 2017) | CVPR'17 | 30.41 | 0.869 | 24.85 | 0.683 | 27.63 | 0.776 | - | - |
| NLEDN (Li et al., 2018a) | MM'18 | 36.24 | 0.958 | 27.02 | 0.723 | 31.63 | 0.841 | - | - |
| RESCAN (Li et al., 2018b) | ECCV'18 | 30.99 | 0.887 | 26.70 | 0.683 | 28.85 | 0.785 | 0.15 | 32.32 |
| PreNet (Ren et al., 2019) | CVPR'19 | 36.63 | 0.968 | 26.43 | 0.729 | 31.53 | 0.849 | 0.17 | 66.58 |
| UMRL (Yasarla & Patel, 2019) | CVPR'19 | 35.76 | 0.962 | 25.89 | 0.726 | 30.83 | 0.844 | 0.98 | 16.50 |
| JORDER-E (Yang et al., 2019) | TPAMI'19 | 33.65 | 0.925 | 26.56 | 0.713 | 30.11 | 0.819 | 4.21 | 273.68 |
| MSPFN (Jiang et al., 2020) | CVPR'20 | 38.61 | 0.975 | 26.45 | 0.727 | 32.53 | 0.851 | 21.00 | 708.44 |
| CCN (Quan et al., 2021) | CVPR'21 | 39.17 | 0.981 | 27.46 | 0.737 | 33.32 | 0.859 | 3.75 | 245.85 |
| MPRNet (Zamir et al., 2021) | CVPR'21 | 40.81 | 0.981 | 27.29 | 0.736 | 34.05 | 0.859 | 3.64 | 148.55 |
| DGUNet (Mou et al., 2022) | CVPR'22 | 41.09 | 0.983 | 27.52 | 0.737 | 34.31 | 0.860 | 12.18 | 199.74 |
| Uformer (Wang et al., 2022) | CVPR'22 | 40.69 | 0.972 | 26.89 | 0.730 | 33.79 | 0.851 | 20.63 | 43.86 |
| Restormer (Zamir et al., 2022) | CVPR'22 | 41.42 | 0.980 | 27.39 | 0.742 | 34.41 | 0.861 | 26.12 | 140.99 |
| IDT (Xiao et al., 2022) | TPAMI'22 | 41.61 | 0.983 | 27.51 | 0.743 | 34.56 | 0.863 | 16.41 | 61.90 |
| NAFNet (Chen et al., 2022a) | ECCV'22 | 40.39 | 0.972 | 27.49 | 0.729 | 33.94 | 0.851 | 40.60 | 16.19 |
| UDR-S2Former (Chen et al., 2023a) | ICCV'23 | 42.39 | 0.988 | **27.90** | **0.745** | 35.15 | 0.867 | 8.53 | 21.58 |
| UDR-Mixer-L | Ours | **44.31** | **0.992** | 27.12 | 0.743 | **35.72** | **0.868** | 2.60 | 64.65 |

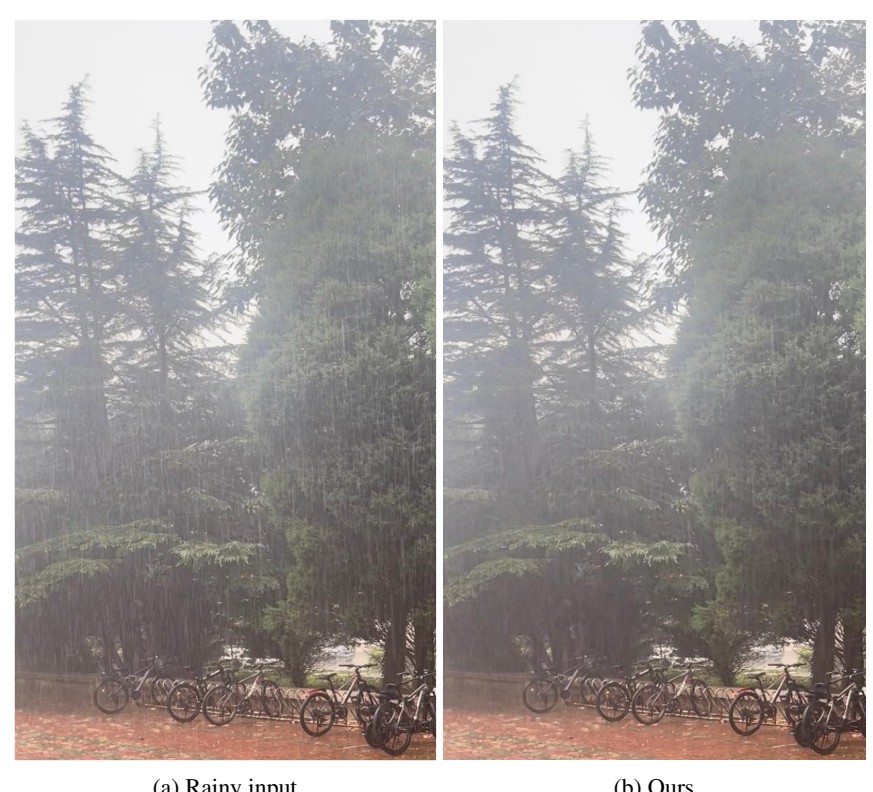

(a) Rainy input                      (b) Ours

Figure 14: Failure case.

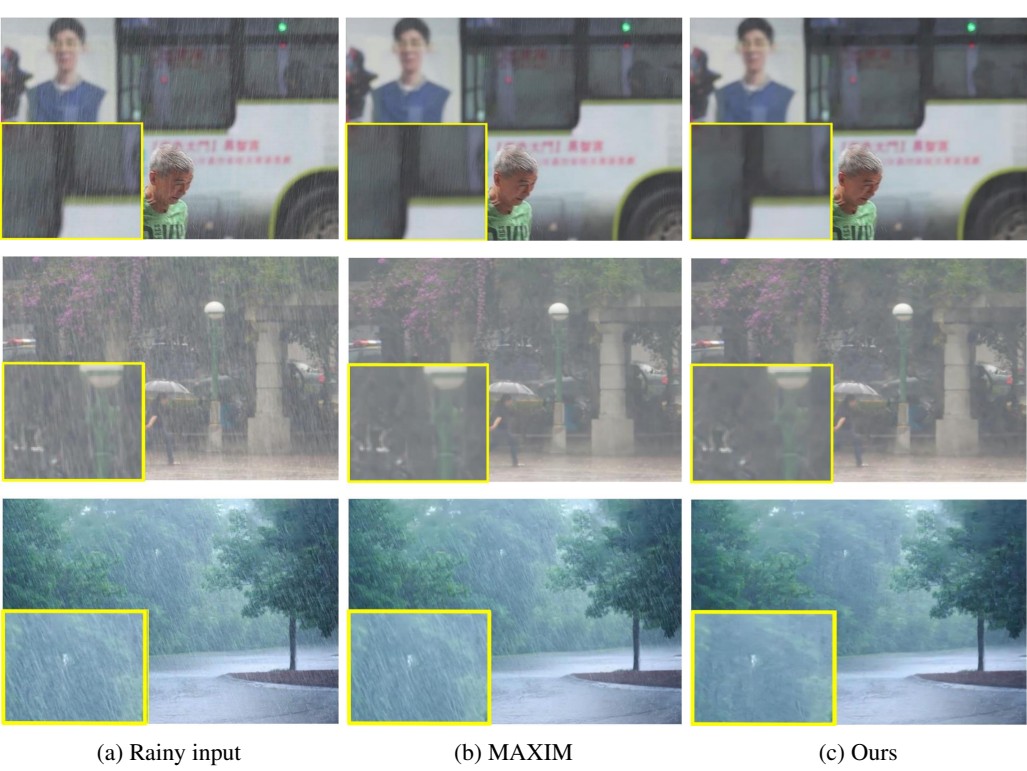

(a) Rainy input           (b) MAXIM           (c) Ours

Figure 15: Comparison results with MAXIM (Tu et al., 2022).

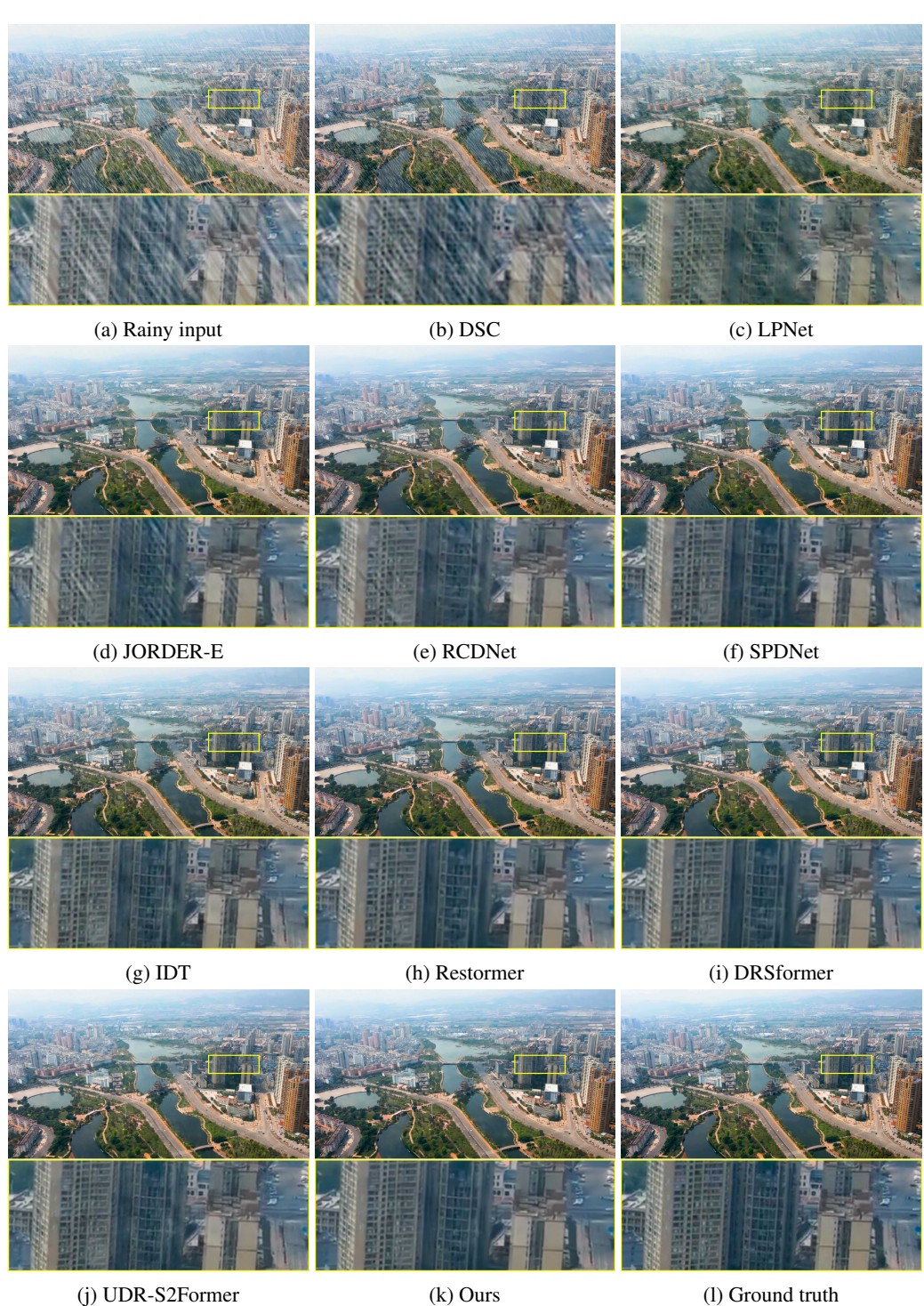

Figure 16: Visual quality comparison on the 4K-Rain13k dataset. Compared with the derained results in (b)-(j), our method recovers a high-quality image with clearer details. Zooming in the figures offers a better view at the deraining capability.

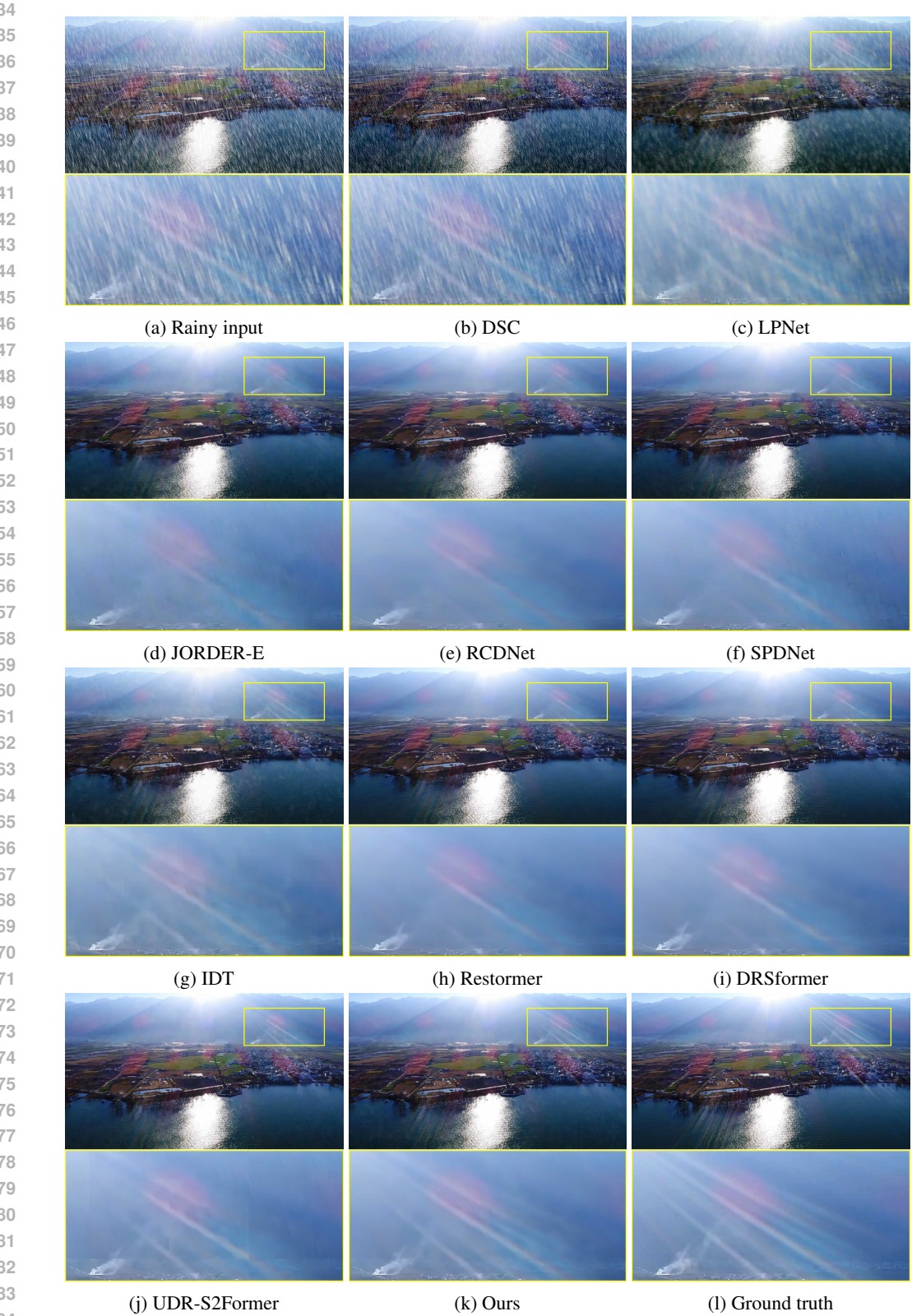

Figure 17: Visual quality comparison on the 4K-Rain13k dataset. Compared with the derained results in (b)-(j), our method recovers a high-quality image with clearer details. Zooming in the figures offers a better view at the deraining capability.

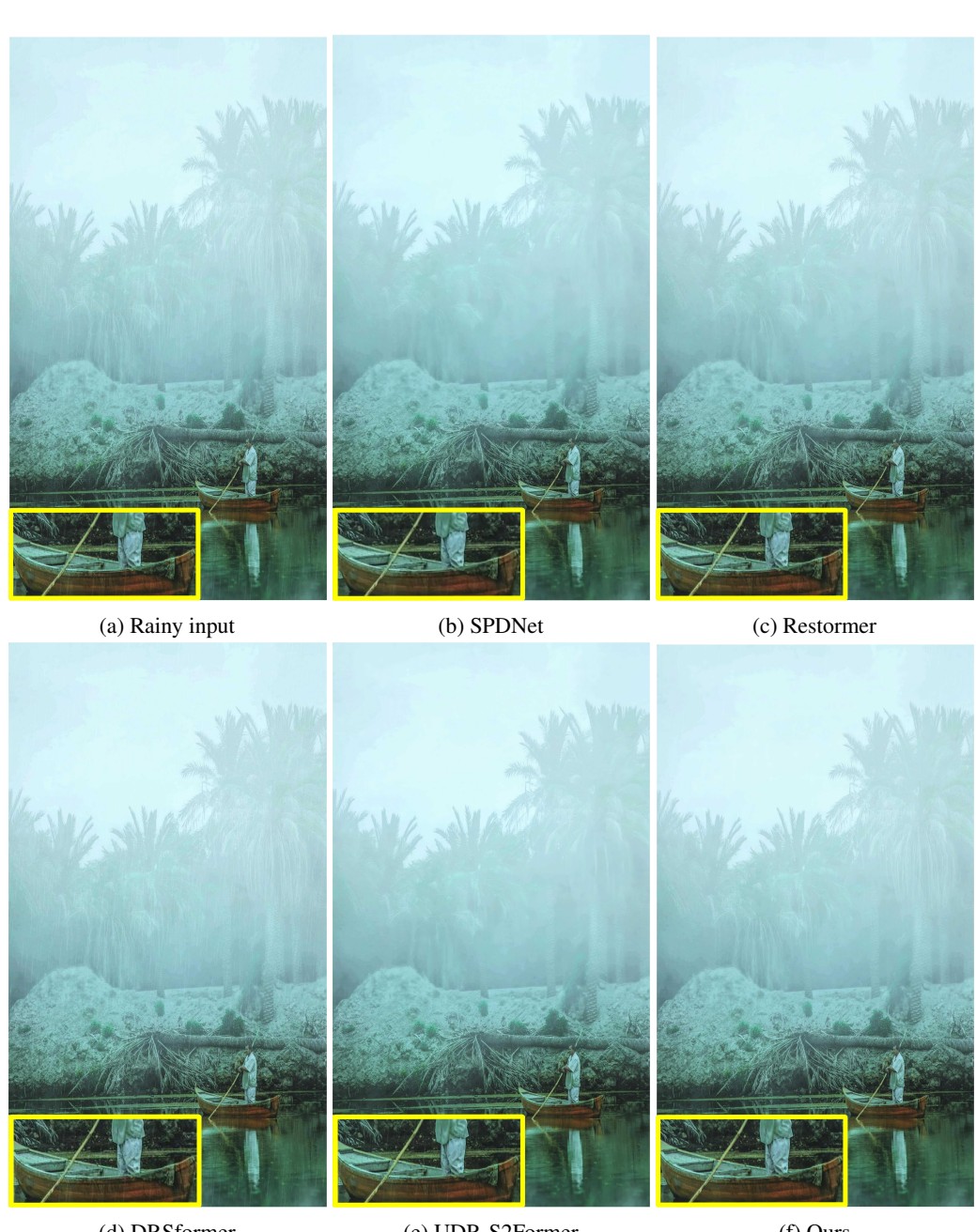

(a) Rainy input      (b) SPDNet      (c) Restormer

(d) DRSformer      (e) UDR-S2Former      (f) Ours

Figure 18: Visual quality comparison on a real-world UHD rainy image from the collected 4K-RealRain. Compared with the derained results in (b)-(e), our method recovers a clearer image. Zooming in the figures offers a better view at the deraining capability.

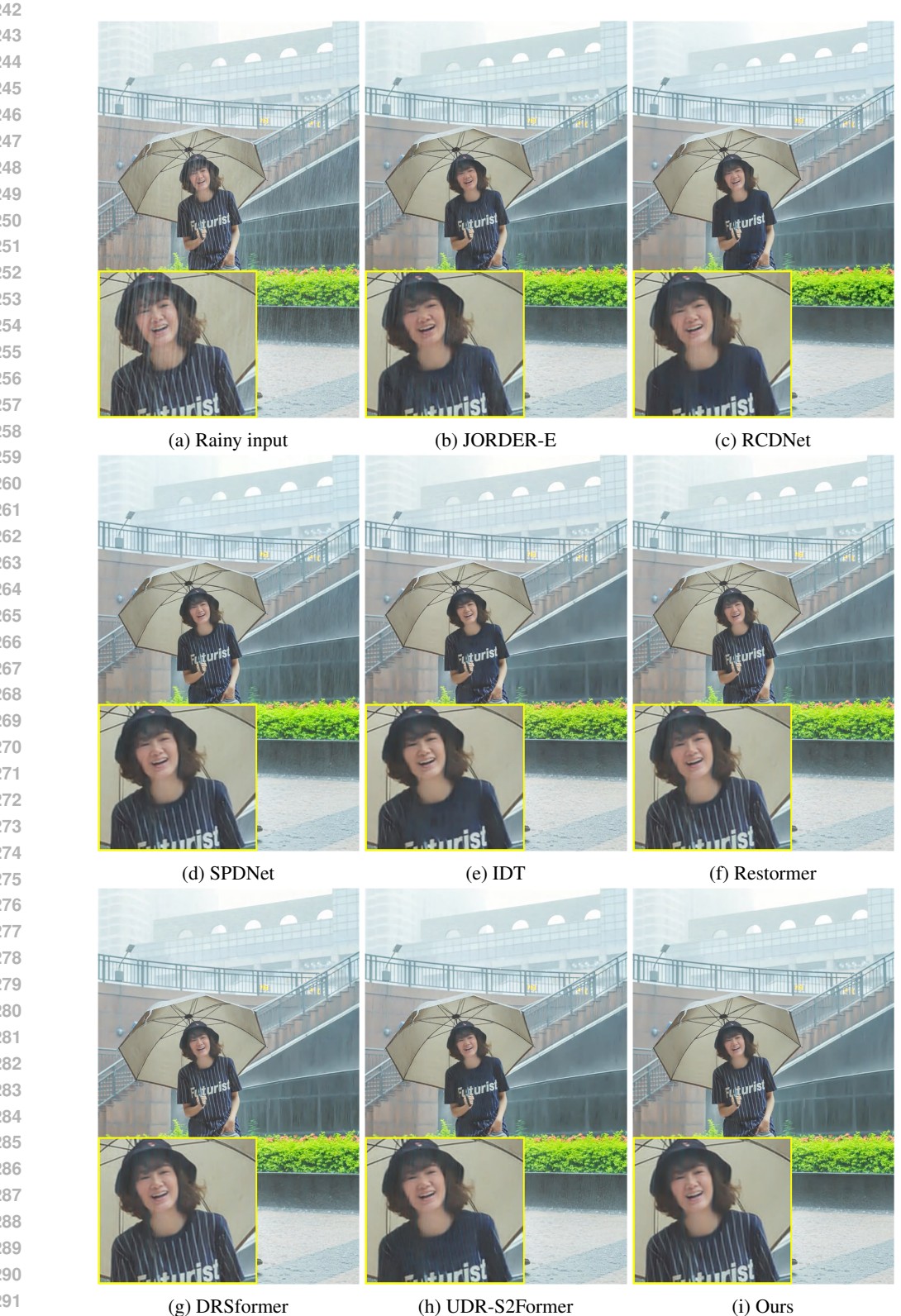

Figure 19: Visual quality comparison on a real low-resolution rainy image from the RE-RAIN dataset. Compared with the derained results in (b)-(h), our method recovers a clearer image. Zooming in the figures offers a better view at the deraining capability.

