# OpenReview forum: "Towards Ultra-High-Definition Image Deraining: A Benchmark and An Efficient Method"
_ICLR.cc/2025/Conference — Submitted to ICLR 2025_

### Official Review · Reviewer_nsEG · 2024-10-21

**Soundness:** 2
**Presentation:** 3
**Contribution:** 3
**Rating:** 5
**Confidence:** 5

**Summary:**

This paper focuses on ultra-high-definition (UHD) image deraining and introduces the first UHD image deraining dataset, 4K-Rain13k. The authors also propose a dual-branch network architecture, UDR-Mixer, to better address this task. Specifically, a spatial feature rearrangement layer is employed to capture long-range information in UHD images, while a frequency feature modulation layer complements the reconstruction of UHD image details. Qualitative and quantitative experimental results demonstrate that the proposed method outperforms existing state-of-the-art approaches on both the proposed dataset and several public datasets.

**Strengths:**

1. The proposed dataset could benefit the community and inspire further research.
2. The experiments and ablation studies are comprehensive, demonstrating the effectiveness of the proposed method compared to existing state-of-the-art approaches.

**Weaknesses:**

1. The technical contribution of the proposed method is limited. The proposed Spatial Feature Rearrangement Layer (SFRL) is quite similar to the Global Feature Modulation Layer (GFML) in MixNet [1].  The author should explicitly compare and contrast the two modules. Adding a paragraph to discuss the key similarities and differences, and highlighting the novel design of the proposed SFRL, is essential.
2. Many implementation details are unclear and not described in the paper. For instance, the authors mention that raindrop generation is modeled as a motion blur process to synthesize corresponding raindrop images. However, how exactly are these images generated? Additionally, the authors claim that alpha blending is used to ensure fidelity, but how are the blending weights determined, and how do different weights affect fidelity?  The authors should include a more detailed subsection on the dataset generation process, covering the exact motion blur modeling and alpha blending techniques used. To provide a clearer understanding, examples showing how different blending weights affect the fidelity of the final synthesized images should also be provided.
3. To better demonstrate the efficiency of the proposed method, the comparison of runtime should be reported. The author should include a table or figure comparing the runtime of the proposed approach to other SOTA methods on a specific hardware setup.

[1] Wu et al., MixNet: Efficient Global Modeling for Ultra-High-Definition Image Restoration. arXiv2024.

**Questions:**

The authors should address the issues mentioned in the weaknesses, particularly those related to the technical design and implementation details.

---

> ### Author Response · Authors · 2024-11-17
>
> We appreciate the efforts and valuable suggestions provided by the reviewer. We will address the concerns outlined below:
>
> **[W1]**
>
> According to the reviewer's suggestion, we have added a section to discuss with the closely-related method (see Appendix A.6). Compared to the Global Feature Modulation Layer (GFML) in MixNet, our SFRL has several main differences: (1) Flexible Rearrangement: Unlike the fixed swapping of adjacent elements in GFML, SFRL rearranges the positions of all elements in the 3D feature map during each dimension transformation. This allows our method to flexibly capture global spatial information. (2) Multi-Scale Integration: While GFML operates at a single scale, SFRL is embedded within a multi-scale encoder-decoder network backbone, enabling more effective exploration of multi-scale information, which is crucial for image deraining. (3) Enhanced UHD Restoration: As discussed in Section 4.3 of the main paper, MixNet directly downsamples UHD images to learn spatial features, leading to detail loss. Our method addresses this issue by introducing the Frequency-aware Feature Modulation Layer (FFML), which enhances UHD image restoration quality through joint learning in both spatial and frequency information.
>
> **[W2]**
>
> Thank you for the reviewer's suggestions. We have added more implementation details for dataset construction in the revised version (see Appendix A.1).
>
> We model the generation of rain streak layers as a motion blur process, leveraging two essential characteristics of rain streaks: their repeatability and directionality. Formally, this can be represented as:
> $$
> \mathbf{S}=\mathbf{K}(l, \theta, w) * \mathbf{N}(n)
> $$
> where $\mathbf{N}$ represents the rain mask derived from random noise $n$, where we utilize uniform random numbers combined with thresholds to adjust the noise level. The parameters $l$ and $\theta$ correspond to the length and angle of the motion blur kernel $\mathbf{K} \in \mathbb{R}^{p \times p}$. Additionally, a rotated diagonal kernel is applied with Gaussian blur to control the rain streak thickness $w$. The values for noise quantity $n$, rain length $l$, rain angle $\theta$, and rain thickness $w$ are sampled from the ranges $[100,500]$, $[10,40]$, $[0^{\circ},60^{\circ}]$, and $[3,7]$, respectively. The symbol $*$ denotes the spatial convolution operator.
>
>
> We utilize alpha blending to combine the rain layer with the background layer. Here, the alpha value of each pixel in a layer determines the extent to which colors from underlying layers are visible through the current layer's color. Mathematically, this process is expressed as:
> $$
> \mathbf{R}_{r, g, b}=\alpha \odot \mathbf{S}+(1-\alpha) \odot \mathbf{B}
> $$
> where $\alpha$ represents the blending ratio, which is empirically set within the range $[0.8, 0.9]$. The symbol $\odot$ indicates element-wise multiplication. The $R$, $G$, and $B$ channels are handled independently during the process. We show an example of synthesizing rainy images with different blending ratios in Figure 8 in Appendix A.1. It is evident that when the blending ratios are set to 0.8 and 0.9, the rain streaks become invisible in the white sky region, resulting in greater harmony with real rainy images.
>
>
>
> **[W3]**
>
> According to the reviewer's suggestion, we calculate the runtime of our method and recent image deraining methods on 4K images. We conduct on a machine equipped with an NVIDIA RTX 4090 GPU. The table below shows that our method achieves lower inference time. We have added this result in the revised paper.
>
> |       Methods        | Runtime (s) |
> | :------------------: | :---------: |
> |         IDT          |   0.1677    |
> |      Restormer       |   0.6324    |
> |      DRSformer       |   1.2061    |
> |     UDR-S2Former     |   0.8194    |
> | **UDR-Mixer (Ours)** | **0.0072**  |
> |                      |             |

---

> > ### Author Response · Authors · 2024-11-29
> >
> > Dear Reviewer nsEG,
> >
> > We greatly appreciate the time and effort you dedicated to reviewing our paper. As the deadline for the discussion period approaches and we have yet to receive your feedback, we kindly request that you share any remaining concerns. Please let us know if we can provide any additional information or clarification.
> >
> > Thank you once again for your contributions to the development of our paper.
> >
> > Authors of Submission 5779

---

### Official Review · Reviewer_nJWL · 2024-10-23

**Soundness:** 3
**Presentation:** 2
**Contribution:** 3
**Rating:** 6
**Confidence:** 3

**Summary:**

The paper introduces a 4K dataset with synthetically generated rain streaks, for the purpose of data-driven de-raining image enhancement. The dataset contains 13K images and is produced by taking some of the specific challenges in synthesizing rain at this resolution level into account. An MLP-Mixer-based network is formulated, with the aim to allow for de-raining of UHD images with less computational demands compared to previous state-of-the-art methods. The architecture is designed using separate branches for spatial and frequency domain processing. Low-level image features are processed in an autoencoder network composed of spatial feature mixing blocks, while an auxiliary branch performs frequency domain mixing to promote quality at the UHD resolution. Experiments compare against previous methods that are applicable in the UHD domain, on the synthetic images with reference-based metrics and on real rainy images using non-reference quality metrics. Additional experiments also compare on datasets of lower resolution, where the proposed method is adapted to work better at this resolution. An ablation study explore the different components of the network, e.g., different strategies for mixing.

**Strengths:**

The focus on UHD image de-raining is a natural extension of previous work, and there is good value in providing a dataset for this purpose. The proposed design focusing on efficient processing at the UHD resolution level is demonstrated to be both efficient and generating competitive performance in comparison to previous methods. Although this is an adaptation of previous work, there seem to be some novel aspects in terms of promoting detail reconstruction at high resolution. Overall, the combination of UHD focus, providing a dataset for this purpose, and competitive results, makes for relatively high significance.

**Weaknesses:**

One of the formulated contributions is the dataset. However, in its current form the paper falls short of demonstrating this as a strong contribution. There is very little detail regarding the dataset. Online sources are not specified, and the simulation of rain is under-explained. The motivation for why the simulation is different in UHD images is unclear and speculative, and there is very little information on the transformations used to account for this in the simulation. It would not be possible to reproduce the simulation without a significant amount of additional details.

While the paper has an ok structure overall, the writing could be improved in terms of formulations and grammar. Also, the reference format needs to be revised (differentiating between \citet and \citep depending on usage).

**Questions:**

* The comparison to MAXIM is only performed in terms of one example image. How does the proposed model compare in terms of PSNR on the Rain13k dataset?
* How was the selection of previous methods done? Specifically, how did you determine that a model is not possible to use on the UHD dataset? Is it due to architectural constraints, memory consumption, or computational complexity? If memory or computations is a deciding factor, what is the threshold for deciding if it is not applicable to UHD?
* In relation to training of the compared methods, it is explained that "We uniformly select the weights from their final training epoch for testing purposes". Does this mean that all models, including the proposed, are trained for an equal number of epochs?
* How was other hyper-parameters tuned for the different models? Are all methods trained using their respective default settings? Wouldn't it potentially be other optimal hyper-parameters when training on the UHD dataset?
* Some limitations are mentioned in the end of the paper. It would be interesting to see some examples of typical failure cases. Are there some other notable artifacts in specific situations?

---

> ### Author Response · Authors · 2024-11-17
>
> We appreciate the efforts and valuable suggestions provided by the reviewer. We will address the concerns outlined below:
>
> **[W1]**
>
> Thank you for the reviewer's suggestions. We have added more implementation details for dataset construction in the revised version (see Appendix A.1). We will open-source the code for our data synthesis process to serve as a reference for the community.
>
> The motivation for introducing geometric transformation operation comes from real-world observations. In the physical world, rain streaks are naturally fine and elongated, reflecting their slender geometry and high-speed movement. In high-resolution 4K images, with the increased pixel density and more detailed spatial representation, these characteristics can be captured more accurately. Each rain-affected region is represented with greater precision, allowing the streaks to appear longer and thinner, consistent with their real-world appearance. Conversely, in low-resolution images, the reduced pixel information limits the ability to preserve these fine details. Rain streaks that are physically the same length may appear shorter and thicker due to the loss of spatial resolution and the effects of blurring. This results in a coarser and more ambiguous depiction of rain streaks, where fine structures and elongated shapes are less distinguishable. This disparity highlights the importance of resolution in faithfully representing rain streaks and motivates the development of high-resolution synthesis approaches.
>
> We first generate an initial rain streak layer, then perform geometric transformation using scaling operations, and finally use alpha blending techniques to synthesize the rainy images. Specifically, we model the generation of rain streak layers as a motion blur process, leveraging two essential characteristics of rain streaks: their repeatability and directionality. Formally, this can be represented as:
> $$
> \mathbf{S}=\mathbf{K}(l, \theta, w) * \mathbf{N}(n)
> $$
> where $\mathbf{N}$ represents the rain mask derived from random noise $n$, where we utilize uniform random numbers combined with thresholds to adjust the noise level. The parameters $l$ and $\theta$ correspond to the length and angle of the motion blur kernel $\mathbf{K} \in \mathbb{R}^{p \times p}$. Additionally, a rotated diagonal kernel is applied with Gaussian blur to control the rain streak thickness $w$. The values for noise quantity $n$, rain length $l$, rain angle $\theta$, and rain thickness $w$ are sampled from the ranges $[100,500]$, $[10,40]$, $[0^{\circ},60^{\circ}]$, and $[3,7]$, respectively. The symbol $*$ denotes the spatial convolution operator.
>
>
> We utilize alpha blending to combine the rain layer with the background layer. Here, the alpha value of each pixel in a layer determines the extent to which colors from underlying layers are visible through the current layer's color. Mathematically, this process is expressed as:
> $$
> \mathbf{R}_{r, g, b}=\alpha \odot \mathbf{S}+(1-\alpha) \odot \mathbf{B}
> $$
> where $\alpha$ represents the blending ratio, which is empirically set within the range $[0.8, 0.9]$. The symbol $\odot$ indicates element-wise multiplication. The $R$, $G$, and $B$ channels are handled independently during the process. We show an example of synthesizing rainy images with different blending ratios in Figure 8 in Appendix A.1. It is evident that when the blending ratios are set to 0.8 and 0.9, the rain streaks become invisible in the white sky region, resulting in greater harmony with real rainy images.
>
> **[W2]**
>
> Thank you to the reviewer for pointing out the issue with the reference format. We have updated the reference format in the revised version.

---

> ### Author Response · Authors · 2024-11-17
>
> **[Q1]**
>
> The visual examples compared with MAXIM are intended to evaluate deraining performance on real rainy scenes, rather than on the low-resolution Rain13k dataset. Here, we further provide the quantitative results of our method and MAXIM on real-world image deraining in Table 6. Furthermore, more visual examples are included in Figure 15. These quantitative and qualitative results demonstrate that our method exhibits better generalization capability to real-world scenarios. We have added these results in the revised version.
>
> **[Q2]**
>
> As stated on L322-323, for all deep models, we use an NVIDIA TESLA V100 GPU with 32GB memory to test UHD images. Our experiments reveal that methods such as JORDER-E, RCDNet, SPDNet, Restormer, and DRSformer are unable to infer full-resolution results on UHD images. This limitation arises from their high memory consumption and computational complexity. Due to the diverse architectures of deep learning networks, it is challenging to define a precise threshold for determining whether a model is unsuitable for UHD inference. For models that cannot directly process UHD images, we can adopt a splitting-and-stitching strategy to overcome this limitation, enabling them to handle ultra-high-resolution inputs.
>
> **[Q3]**
>
> Yes. For fair comparison, all models are trained for an equal number of epochs, and we select the weights from their final training epoch for testing purposes. We have clarified this in the revised version.
>
> **[Q4]**
>
> All methods are used with their respective default settings. Following previous UHD studies [1,2], we do not adjust the parameters of the existing models to ensure a fair comparison. We have clarified this in the revised version.
>
> [1] Zheng et al. Ultra-High-Definition Image Dehazing via Multi-Guided Bilateral Learning, CVPR 2021
>
> [2] Li et al. Embedding Fourier for Ultra-High-Definition Low-Light Image Enhancement, ICLR 2023
>
> **[Q5]**
>
> According to the reviewer's suggestion, we have provided a failure case in Appendix A.10. As shown, our method is able to effectively remove rain streaks but fails to handle the presence of fog-like rain accumulation in real rainy scenes. In addition, the splashing effect of raind on the ground is also worth paying attention to in future work, which is crucial for downstream autonomous driving.

---

> > ### Comment · Reviewer_nJWL · 2024-11-26
> >
> > Thanks for the extensive elaboration on the different weaknesses and questions. I appreciate the additional material provided in the paper, and especially in the Appendix. In relation to this, and by reading the other reviews and comments, I will maintain my positive score for the paper.

---

> > > ### Author Response · Authors · 2024-11-29
> > >
> > > Thank you for your acknowledgment of our work and responses. We'll carefully revise our final paper. Your positive rating means a lot to us. We appreciate your constructive feedback that has helped refine our research.

---

### Official Review · Reviewer_rmy5 · 2024-10-29

**Soundness:** 3
**Presentation:** 3
**Contribution:** 3
**Rating:** 8
**Confidence:** 5

**Summary:**

The paper contributes the first large-scale UHD image deraining dataset and proposes MLP-based architecture (UDR-Mixer) to achieve this task. Extensive experiments demonstrate that UDR-Mixer performs favorably against the state-of-the-art approaches.

**Strengths:**

1. The paper constructs the first high-quality UHD image deraining dataset (4K-Rain13k).
2. The paper develops a dual-branch architecture UDR-Mixer, where the spatial feature mixing block and the frequency feature mixing block are proposed.
3. Experimental results demonstrate that UDR-Mixer achieves a favorable trade-off between performance and model complexity.

**Weaknesses:**

1. It is more appropriate to calculate #FLOPs on 4K images.
2. The testing time on 4K images can also be given.
3. For Lines 362-363, why not evaluate by cropping the image into multiple patches?
4. In Table 3, the authors can use some more advanced non-reference IQA metrics, e.g., MANIQA, MUSIQ, and CLIPIQA.
5. It is better to give some 4K deraining results of UDR-Mixer trained on low-resolution datasets.

**Questions:**

Please see 'Weaknesses'.

---

> ### Author Response · Authors · 2024-11-17
>
> We appreciate the efforts and valuable suggestions provided by the reviewer. We will address the concerns outlined below:
>
> **[W1]**
>
> According to the reviewer's suggestion, we calculate the #FLOPs on 4K images. The table below shows that our method achieves lower FLOPs.
>
> |       Methods        | FLOPs (G) |
> | :------------------: | :-------: |
> |      Restormer       |   4900    |
> |      DRSformer       |   5200    |
> |     UDR-S2Former     |   3228    |
> | **UDR-Mixer (Ours)** | **1558**  |
>
> **[W2]**
>
> According to the reviewer's suggestion, we calculate the runtime of our method and recent image deraining methods on 4K images. We conduct on a machine equipped with an NVIDIA RTX 4090 GPU. The table below shows that our method achieves lower inference time. We have added this result in the revised paper.
>
> |       Methods        | Runtime (s) |
> | :------------------: | :---------: |
> |         IDT          |   0.1677    |
> |      Restormer       |   0.6324    |
> |      DRSformer       |   1.2061    |
> |     UDR-S2Former     |   0.8194    |
> | **UDR-Mixer (Ours)** | **0.0072**  |
> |                      |             |
>
> **[W3]**
>
> For some methods (JORDER-E, RCDNet, SPDNet, Restormer, and DRSformer) that are unable to infer full-resolution results on UHD images, existing UHD studies [1,2] typically adopt two strategies: (1) downsample the input to the largest size that the model can handle and then resize the result to the original resolution, (2) split the input into multiple patches and then stitch the result. Researchers in [1] demonstrate that employing the second strategy often leads to undesired boundary artifacts in the output images, as it does not account for the complete structure of the image. Thus, we adopt the first strategy in the original manuscript.
>
> According to the reviewer's suggestion, we further adopt the splitting-and-stitching strategy to evaluate the deraining performance. The table below shows that our method still achieves the best quantitative results, thanks to its ability to perform direct inference on full UHD images. We have updated these results in the revised paper.
>
> |       Methods        | PSNR      |    SSIM    |    MSE    |
> | :------------------: | --------- | :--------: | :-------: |
> |        LPNet         | 27.86     |   0.8924   |  171.33   |
> |       JORDER-E       | 31.01     |   0.9119   |  104.30   |
> |        RCDNet        | 31.41     |   0.9215   |   95.01   |
> |        SPDNet        | 32.38     |   0.9233   |   77.49   |
> |         IDT          | 32.91     |   0.9479   |   57.04   |
> |      Restormer       | 33.70     |   0.9344   |   58.98   |
> |      DRSformer       | 33.47     |   0.9329   |   62.91   |
> |     UDR-S2Former     | 33.36     |   0.9458   |   50.69   |
> | **UDR-Mixer (Ours)** | **34.30** | **0.9505** | **42.03** |
>
> [1] Zheng et al. Ultra-High-Definition Image Dehazing via Multi-Guided Bilateral Learning, CVPR 2021
>
> [2] Li et al. Embedding Fourier for Ultra-High-Definition Low-Light Image Enhancement, ICLR 2023
>
>
>
> **[W4]**
>
> According to the reviewer's suggestion, we further employ the advanced non-reference IQA metric MANIQA to evaluate the quality of different deraining images. The table below shows that our method achieves better perceptual quality with higher MANIQA values.
>
> |       Methods        |   MANIQA   |
> | :------------------: | :--------: |
> |         IDT          |   0.4110   |
> |      Restormer       |   0.4108   |
> |      DRSformer       |   0.4091   |
> |     UDR-S2Former     |   0.4088   |
> | **UDR-Mixer (Ours)** | **0.4130** |
>
>
>
> **[W5]**
>
> We compare the performance of our model trained on the proposed dataset and existing low-resolution dataset (Rain13k) and then test on real-world 4K rainy images. We have provided the comparison results in Appendix A.4 of the revised version. It can be observed that the model trained on our 4K-Rain13k performs better on real-world rainy images, indicating that our dataset effectively reduces the domain gap and enables the model to generalize better to real-world scenarios.

---

> ### Comment · Reviewer_rmy5 · 2024-11-22
> **Thanks**
>
> Thanks to the authors for the answer. The authors have addressed my concerns well, and I consider the paper acceptable.

---

> > ### Author Response · Authors · 2024-11-29
> >
> > Thank you for your acknowledgment of our work and responses. We'll carefully revise our final paper. Your positive rating means a lot to us. We appreciate your constructive feedback that has helped refine our research.

---

### Official Review · Reviewer_qtvD · 2024-11-05

**Soundness:** 3
**Presentation:** 3
**Contribution:** 2
**Rating:** 5
**Confidence:** 5

**Summary:**

This paper addresses the task of deraining UHD images, which is not yet well-explored despite advances in low-resolution image deraining research. The authors introduce the first large-scale UHD image deraining dataset, containing 13000 pairs of 4K resolution images. Using this dataset, they benchmark existing methods and identify performance limitations for UHD images. Additionally, the paper presents a model called UDR-Mixer for a balance of both effectiveness and efficiency. Experimental results show that the proposed method outperforms state-of-the-art methods with lower model complexity, making it suitable for practical use.

**Strengths:**

- The paper introduces the first large-scale UHD image deraining dataset, 4K-Rain13k, filling an the gap in high-resolution deraining research
- The method is evaluated on both synthetic and real dataset, which verifies its practical use.

**Weaknesses:**

- I do not really understand how oversized models are evaluated from the expressions in Line 361- 363. If I do not misunderstand, the authors first resize the image to the largest size that can be processed by a single GPU, and then perform these deraining models, finally enlarge the size to the original size and evaluate the metrics. If so, I concern it would be unfair for these compared methods. Changing the resolution or dpi during the inference would greatly reduce the performance, since the model is not trained on this resolution, and resize operation would also lead to blurry artifacts. A more rigorous way is to divide the input image to several overlapped patches, evaluate on each patch, and then combine them.
- I am not sure whether it is necessary to have a training dataset with 4K resolution. We cannot send the full resolution images during training, just like this paper crop $768 \times 768$ patches for training. So from my view, it is equivalant to have a training dataset with $768 \times 768$  resolution. Please explain the rationale behind using 4K resolution for the training dataset given the $768 \times 768$ patch size used in training.
- I concern whether the rain streak generation approach is authentic enough to generate 4K images. With higher resolution, the details of real rain streak (like the texture, position, perspective relationship) should be clearer than that in a low-resolution image. It requires more realistic generation method for high resolution. However, in Fig. 2, the generated rain streak seems like a separate layer in front of the scene, and the discrepencies are enlarged in high resolution images.
- The dataset is regarded as an important contribution to this paper. Therefore, the authors should also conduct ablation studies on dataset, to verify why 4K training dataset instead of low-resolution training dataset is important for 4K deraining testing under synthetic/real scenarios. Specifically, conduct ablation studies comparing models trained on 4K data vs. lower resolution data, but tested on 4K images.

**Questions:**

Please see my comments in the weakness and answer the questions I raised in the weakness.

---

> ### Author Response · Authors · 2024-11-17
>
> We appreciate the efforts and valuable suggestions provided by the reviewer. We will address the concerns outlined below:
>
> **[W1]**
>
> I'm sorry for any confusion caused. For some methods (JORDER-E, RCDNet, SPDNet, Restormer, and DRSformer) that are unable to infer full-resolution results on UHD images, existing UHD studies [1,2] typically adopt two strategies: (1) downsample the input to the largest size that the model can handle and then resize the result to the original resolution, (2) split the input into multiple patches and then stitch the result. Researchers in [1] demonstrate that employing the second strategy often leads to undesired boundary artifacts in the output images, as it does not account for the complete structure of the image. Thus, we adopt the first strategy in the original manuscript.
>
> According to the reviewer's suggestion, we further adopt the splitting-and-stitching strategy to evaluate the deraining performance. The table below shows that our method still achieves the best quantitative results, thanks to its ability to perform direct inference on full UHD images. We have updated these results in the revised paper.
>
> |       Methods        | PSNR      |    SSIM    |    MSE    |
> | :------------------: | --------- | :--------: | :-------: |
> |        LPNet         | 27.86     |   0.8924   |  171.33   |
> |       JORDER-E       | 31.01     |   0.9119   |  104.30   |
> |        RCDNet        | 31.41     |   0.9215   |   95.01   |
> |        SPDNet        | 32.38     |   0.9233   |   77.49   |
> |         IDT          | 32.91     |   0.9479   |   57.04   |
> |      Restormer       | 33.70     |   0.9344   |   58.98   |
> |      DRSformer       | 33.47     |   0.9329   |   62.91   |
> |     UDR-S2Former     | 33.36     |   0.9458   |   50.69   |
> | **UDR-Mixer (Ours)** | **34.30** | **0.9505** | **42.03** |
>
> [1] Zheng et al. Ultra-High-Definition Image Dehazing via Multi-Guided Bilateral Learning, CVPR 2021
>
> [2] Li et al. Embedding Fourier for Ultra-High-Definition Low-Light Image Enhancement, ICLR 2023
>
>
>
> **[W2]**
>
> Firstly, directly training on full 4K images poses significant challenges due to the limited memory capacity of current GPU devices. In image restoration, scaling up the resolution leads to prohibitively high GPU memory consumption, making it infeasible to process such high-resolution data in an end-to-end training framework. Therefore, training on small patches is a widely adopted and practical approach in the field. In our experiment, we choose $768 \times 768$ as the maximum patch size based on GPU memory limitations.
>
> Secondly, although the network is trained on patches, our dataset is constructed using full 4K-resolution images. As stated on L177-180, we fully consider the unique attributes of rain streaks on 4K high-resolution images during the dataset construction process. Training on patches sampled from this dataset enables the network to effectively learn these features, as the rain streak patterns are preserved within the patches.
>
> Thirdly, while the network is trained on patches for computational efficiency, the testing and evaluation are conducted on full 4K images. This ensures that the model's performance is directly assessed on the target resolution, demonstrating its ability to handle UHD image.

---

> ### Author Response · Authors · 2024-11-17
>
> **[W3]**
>
> As stated on L161-186, we find that the geometric inconsistency in the synthesis of low resolution and high-resolution rainy images, with noticeable discrepancies in the length and thickness of rain streaks. Conventional methods may struggle to preserve realistic attributes such as texture, position, and perspective relationships at higher resolutions. To overcome this, our geometric transformation approach explicitly considers these factors, ensuring that the generated rain streaks align naturally with the scene and exhibit proper perspective and scaling characteristics.
>
> As stated on L159, to mitigate the appearance of rain streaks as a separate layer, we incorporate alpha blending into the synthesis process. This blending technique ensures a smooth integration of rain streaks with the background, preserving the physical realism of rain’s interaction with the scene (e.g., partial occlusion, transparency effects). We have added more details for dataset construction in the revised version (see Appendix A.1).
>
> In Appendix A.3, we have provided a quantitative analysis of the proposed dataset. Specifically, we adopt the Kullback-Leibler Divergence (KLD), also known as relative entropy, to measure the difference between two probability distributions (i.e., synthetic image and real-world image). Figure 10 presents the comparison results of the representative synthetic benchmarks and our benchmark, showing that our 4K-Rain13k is close to the distribution of real-world rainy images. The reason behind this is that 4K-Rain13k fully considers the realism and diversity of the rain patterns, thereby narrowing the domain gap between synthetic and real images.
>
> **[W4]**
>
> Thank you for the reviewer's suggestions. We compare the performance of our model trained on the proposed dataset and existing low-resolution dataset (Rain13k) and then test on real-world 4K rainy images. We have provided the comparison results in Appendix A.4 of the revised version. It can be observed that the model trained on our 4K-Rain13k performs better on real-world rainy images, indicating that our dataset effectively reduces the domain gap and enables the model to generalize better to real-world scenarios.

---

> > ### Comment · Reviewer_qtvD · 2024-11-25
> >
> > Thanks for addressing my concerns. Some of my concerns are solved. Although we still do not know how much performance degradations are caused by the splitting-stitching operation and whether these methods with such operations could surpass the proposed method if they can directly evaluate a UHD image, I do agree that the new version is better than the previous resize-enlarge operation. As for the rain generation, I still think the use of 4K resolution enlarges the unrealism of rain. I mean when synthesizing on low resolution images, the requirement for rain realism is reduced since we do not need very fine-grained details in low resolution, but  it's not the case in UHD image. We can easily tell the synthetic rains from real rains in UHD image, just like Fig. 9 in the appendix.

---

> > > ### Author Response · Authors · 2024-11-26
> > >
> > > Thanks for the reviewers’ feedback. We sincerely appreciate the reviewers' positive recognition of the modifications in our revised version. Below, we provide further responses to the two points raised by the reviewers:
> > >
> > > [Q1]
> > >
> > > For the comparison methods, both IDT and UDR-S2Former can directly evaluate UHD images due to their inherent splitting-stitching operations. The above experimental results demonstrate that our method achieves superior performance. Moreover, it is worth mentioning that our approach also achieves lower model complexity, which is crucial for the practical application of UHD image processing. In summary, our method provides an effective new solution for image deraining, achieving a better trade-off between efficiency and performance.
> > >
> > > [Q2]
> > >
> > > We understand the reviewer’s perspective that higher resolutions require more attention to the fine-grained details of rain to ensure realism. In this paper, we incorporate geometric transformations and alpha blending into the synthesis process to adjust the realism of the synthesized rain. The goal of our method is to reduce the domain gap between the synthesized images and real rain images. However, we acknowledge that there is still a discrepancy between the synthesized rain and real rain in high-resolution images. Therefore, improving the realism of UHD rain synthesis remains an area that we plan to explore further in our future work. Our work is the first to introduce the new research problem of UHD image deraining, aiming to advance the field. We believe that this work provides the community with a fresh research perspective, rather than continuing to focus on previous low-resolution rain datasets. We would like to thank the reviewers once again for their valuable suggestions, which offer important guidance for the future development of this field.

---

### Author Response · Authors · 2024-12-03
**A Thank Letter and Final Summary**

Dear AC and Reviewers,

We thank all reviewers and area chairs for their valuable time and comments. The reviewer-author discussion deadline set by ICLR is drawing to a close. After discussing with reviewers and providing more clarifications/results/analyses, we would like to give a brief response.

Reviewer rmy5, Reviewer nJWL all hold a positive side for our work.

Reviewer qtvD contributed to the discussion, and we have also provided updated explanations. Hopefully our further discussion will address the reviewer's concerns.

Reviewer nsEG has not responded as of now. Notably, their concerns regarding the dataset implementation details (W2) and runtime (W3) overlap with those raised by other reviewers (Reviewer rmy5 - W2 and Reviewer nJWL - W1). These problems have been addressed in the revised version, as highlighted by Reviewer nJWL, who commented: *"Thanks for the extensive elaboration on the different weaknesses and questions. I appreciate the additional material provided in the paper, and especially in the Appendix"*.

We believe it is professional to participate in the discussion before the ending date and justify the rating.

Once again we thank all reviewers and area chairs!

Best,

Paper 5779 Authors.

---

### Meta-Review · Area_Chair_qwFx · 2024-12-25

**Metareview:**

This paper presents the first dataset for UHD-resolution image draining and a computationally efficient deraining model for such high-resolution images. The dataset outstands the existing lower-resolution datasets as the rain streak shape patterns differ in higher resolutions. The existing methods and the proposed method are compared in both the high- and low-resolution datasets.

* However, there is a concern about the technical novelty in the proposed method.

The proposed SFRL is quite similar to GFML; multi-scale integration compared with single-scale GFML is not very significant.
Another proposed component, FFML, is without technical justification for why FFT is employed.
Lack of significant differences and the lack of design justification make it hard to say the contribution is novel and technically effective even if the experimental results are better.

* The reviewers asked for the details how the compared methods were trained and how were the hyper-parameters were chosen. The authors said that all methods were with the default settings, however, the revised paper does not appear so.

* As the dataset is the major contribution of this work, there was a request for grounds why 4K training dataset is necessary compared with low-resolution training data validated on 4K test environment with real/synthetic scenarios. The authors responded by suggesting KL divergence between the proposed dataset and real rainy images. The expectation is not fulfilled.

Considering the reviews and the author responses, the authors are suggested to improve the work.

**Additional Comments On Reviewer Discussion:**

While several questions from the reviewers are answered, there remain several unresolved concerns.

The technical novelty of the proposed deraining model design, details of how the compared methods are trained, lack of analysis to justify the need for 4K training dataset are those.
The authors are advised to improve these aspects.

---

### Decision · Program_Chairs · 2025-01-22

Reject